# Postural adaptations may contribute to the unique locomotor energetics seen in hopping kangaroos

Lauren Thornton[1], Taylor Dick[2], John R Hutchinson[3], Glen A Lichtwark[4], Craig P McGowan[5], Jonas Rubenson[6], Alexis Wiktorowicz-Conroy[3], Christofer J Clemente[1,2]*

[1]School of Science, Technology, and Engineering, University of the Sunshine Coast, Sippy Downs, Australia; [2]School of Biomedical Sciences, University of Queensland, St Lucia, Australia; [3]Structure and Motion Lab, Department of Comparative Biomedical Sciences, The Royal Veterinary College, Hatfield, United Kingdom; [4]School of Exercise and Nutrition Science, Queensland University of Technology, Brisbane, Australia; [5]Department of Integrative Anatomical Sciences, Keck School of Medicine University of Southern California, Los Angeles, United States; [6]Biomechanics Laboratory, Department of Kinesiology, The Pennsylvania State University, University Park, United States

*For correspondence:
cclement@usc.edu.au

Competing interest: The authors declare that no competing interests exist.

## eLife Assessment

This **valuable** biomechanical analysis of kangaroo kinematics and kinetics across a range of hopping speeds and masses is a step towards understanding a long-standing problem in locomotion biomechanics: the mechanism for how kangaroos, unlike other mammals, can increase hopping speed without a concomitant increase in metabolic cost. The authors **convincingly** demonstrate that changes in kangaroo posture with speed increase tendon stress/strain and hence elastic energy storage/return. This greater tendon elastic energy storage/return may counteract the increased cost of generating muscular force at faster speeds and thus allows for the invariance in metabolic cost. This methodologically impressive study sets the stage for further work to investigate the relation of hopping speed to metabolic cost more definitively.

**Abstract** Hopping kangaroos exhibit remarkably little change in their rate of metabolic energy expenditure with locomotor speed compared to other running animals. This phenomenon may be related to greater elastic energy savings due to increasing tendon stress; however, the mechanisms which enable the rise in stress without additional muscle work remain poorly understood. In this study, we created a three-dimensional (3D) kangaroo musculoskeletal model, integrating 3D motion capture and force plate data, to analyse the kinematics and kinetics of hopping red and grey kangaroos. Using our model, we evaluated how body mass and speed influence (i) hindlimb posture, (ii) effective mechanical advantage (EMA), (iii) the associated tendon stress in the ankle extensors, and (iv) ankle work during hopping. We found that increasing ankle dorsiflexion and metatarsophalangeal plantarflexion likely played an important role in decreasing ankle EMA by altering both the muscle and external moment arms, which subsequently increased energy absorption and peak tendon stress at the ankle. Surprisingly, kangaroo hindlimb posture changes appeared to contribute to increased tendon stress, allowing more elastic energy storage at faster speeds. These posture-mediated increases in elastic energy storage and return could be a key factor enabling kangaroos to achieve energetic benefits at

faster hopping speeds, but may limit the performance of large kangaroos due to the risk of tendon rupture.

## Introduction

Kangaroos and other macropods are unique in both their morphology and their locomotor style. At slow speeds, they use a pentapedal gait, where the forelimbs, the hindlimbs, and the tail all contact the ground, while at faster movement speeds, they use their distinctive hopping gait (*Dawson and Taylor, 1973*; *O'Connor et al., 2014*). Their uniqueness extends into their energetics of locomotion. As far back as the 19th century, researchers noticed that the metabolic cost of running in quadrupeds and bipeds, like dogs, horses and humans, increased linearly with speed (*Zuntz, 1897*; *Taylor et al., 1970*; *Heglund et al., 1982*; *Taylor et al., 1982*). To explain why metabolic rate increased at faster running speeds among quadrupeds and bipeds, *Kram and Taylor, 1990* refined the 'cost of generating force' hypothesis (*Taylor et al., 1980*). They reasoned that the decrease in contact time with increased speed, reflects an increase in the rate of generating muscle force, and the rate of cross-bridge cycling (*Kram and Taylor, 1990*). This was supported for a diverse range of running and hopping animals, suggesting that metabolic rate was inversely proportional to contact time (*Kram and Taylor, 1990*). Yet hopping macropods appear to defy this trend. On treadmills, both red kangaroos (~20 kg) and tammar wallabies (~5 kg) showed little to no increase in the rate of oxygen consumption with increased hopping speed (*Dawson and Taylor, 1973*; *Baudinette et al., 1992*; *Kram and Dawson, 1998*). The underlying mechanisms explaining how macropods are able to uncouple hopping speed and energy cost is not completely understood (*Thornton et al., 2022*).

The ability to uncouple speed and energy expenditure in macropods is likely related to the behaviour of their ankle extensor muscle-tendon units, which store and return elastic strain energy (*Morgan et al., 1978*; *Biewener et al., 2004b*; *McGowan et al., 2005*). In tammar wallabies, ankle tendon stress increases with hopping speed, leading to a greater rise in elastic strain energy return than muscle work, which increases the proportion of work done by tendon recoil while muscle work remains near constant (*Baudinette and Biewener, 1998*; *Biewener et al., 1998*). Size-related differences in ankle extensor tendon morphology (*Bennett and Taylor, 1995*; *McGowan et al., 2008*), and the resultant low strain energy return, may explain why small (<3 kg) hopping macropods and rodents appear not to be afforded the energetic benefits observed in larger macropods (*Thompson et al., 1980*; *Biewener et al., 1981*; *Biewener et al., 1998*) (but see *Christensen et al., 2022*). However, tendon morphology alone is insufficient to explain why large macropods can increase speed without cost, while large quadrupeds with similar tendon morphology cannot (*Dawson and Webster, 2010*). The most obvious difference between macropods and other mammals is their hopping gait, but previously proposed mechanisms to explain how hopping could reduce the metabolic cost of generating muscle force, such as near-constant stride frequency (*Heglund and Taylor, 1988*; *Dawson and Webster, 2010*) or respiratory-stride coupling (*Baudinette et al., 1987*), do not distinguish between small and large macropods, nor galloping quadrupeds (*McGowan and Collins, 2018*).

Postural changes are another mechanism which could contribute to reduced energetic costs by altering the leverage of limb muscles. The EMA is the ratio of the internal muscle-tendon moment arm (the perpendicular distance between the muscle's force-generating line-of-action and the joint centre) and the ground reaction force (GRF) moment arm (the perpendicular distance between the vector of the GRF to the joint centre). Smaller EMA requires greater muscle force to produce a given force on the ground, thereby demanding a greater volume of active muscle, and presumably greater metabolic rates, than larger EMA for the same physiology. In humans, an increase in limb flexion and decrease in limb EMA requires a greater volume of active muscle and increases the metabolic cost of running compared to walking (*Biewener et al., 2004a*; *Kipp et al., 2018*; *Allen et al., 2022*). Changes in EMA with speed are not common in quadrupedal mammals (*Biewener, 1989*) but these are not without precedent. Elephants—one of the largest and noticeably most upright animals—do not have discrete gait transitions, but rather continuously and substantially decrease EMA in limb joints when moving faster, which has also been linked to the increase in metabolic cost of locomotion with speed (*Ren et al., 2010*; *Langman et al., 2012*). EMA is more commonly observed to change with size in terrestrial mammals. Smaller animals tend to move in crouched postures with limbs becoming progressively more extended as body mass increases (*Biewener, 1989*). Yet, rather than metabolic cost, this

postural transition appeared to be driven by the need to reduce the size-related increases in tissue stress (*Dick and Clemente, 2017*; *Clemente and Dick, 2023*). As a consequence of transitioning to more upright limb postures, musculoskeletal stresses in terrestrial mammals are independent of body mass (*Biewener, 1989*; *Biewener, 2005*).

Macropods, in contrast, maintain a crouched posture during hopping. *Kram and Dawson, 1998* explored whether kangaroos transitioned to a more extended limb posture with increasing speed as a potential mechanism for their constant metabolic rate, but did not detect a change in EMA at the ankle across speeds from 4.3 to 9.7 ms$^{-1}$. Furthermore, ankle posture and EMA appears to vary weakly (*McGowan et al., 2008*) or not at all with body mass in macropods (*Bennett and Taylor, 1995*; *Snelling et al., 2017*). As a consequence, stress would be expected to increase with both hopping speed and body mass. Large tendon stresses are required for rapid strain energy return (strain energy $\propto$ stress$^2$) (*Biewener and Baudinette, 1995*). Given that tendon recoil plays a pivotal role in hopping gaits, it is perhaps unsurprising that tendon stress approaches the safe limit in larger kangaroos. The ankle extensor tendons in a moderately sized male red kangaroo (46.1 kg) operate with safety factors near two even at slow hopping speeds (3.9 ms$^{-1}$) (*Kram and Dawson, 1998*), far lower than the typical safety factor of four to eight for mammalian tendons (*Ker et al., 1988*). Tendon stresses are also unusually large in smaller kangaroos. Juvenile and adult western grey kangaroos, ranging in mass from 5.8 to 70.5 kg, all hop with gastrocnemius and plantaris tendon safety factors less than two (*Snelling et al., 2017*). Large tendon stresses may not only be a natural consequence

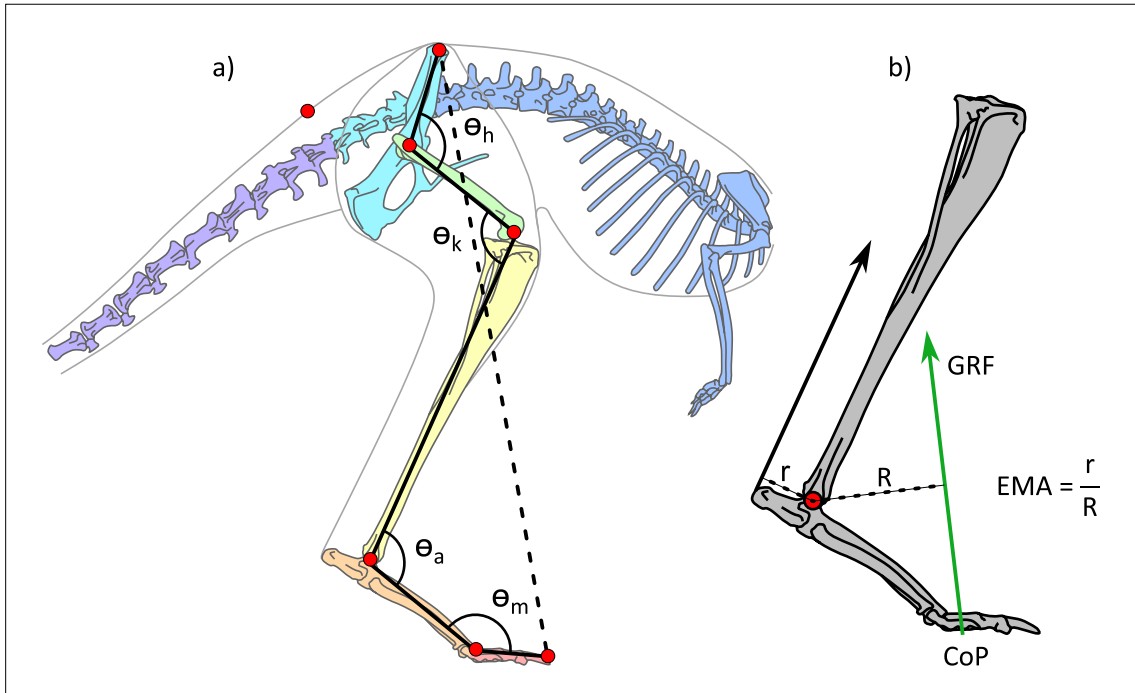

**Figure 1.** Illustration of the kangaroo model and moment arms for the ankle. (**a**) Illustration of the kangaroo model. Total leg length was calculated as the sum of the segment lengths (solid black lines) in the hindlimb and compared to the pelvis-to-toe distance (dashed line) to calculate the crouch factor. Joint angles were determined for the hip, h, knee, k, ankle, a, and metatarsophalangeal, m, joints. The model markers (red circles) indicate the position of the reflective markers placed on the kangaroos in the experimental trials and were used to characterise the movement of segments in the musculoskeletal model. (**b**) Illustration of ankle effective mechanical advantage, EMA, muscle moment arm, r, and external moment arm, R, as the perpendicular distance to the Achilles tendon line of action and ground reaction force (GRF) vector, respectively. The centre of pressure (CoP) was tracked in the fore-aft direction.

The online version of this article includes the following video and figure supplement(s) for figure 1:

**Figure supplement 1.** Distribution of trial speeds and number of trials (**n**) per kangaroo (6.25±5.02 trials per kangaroo).

**Figure 1—video 1.** A red kangaroo hopping on the force plate during data collection.

https://elifesciences.org/articles/96437/figures#fig1video1

**Figure 1—video 2.** Driving the musculoskeletal model with a recorded hopping trial.

https://elifesciences.org/articles/96437/figures#fig1video2

of their crouched posture and tendon morphology, but also be adaptively selected. Considering this, kangaroos may adjust their posture to increase tendon stress and its associated elastic energy return. If so, there would likely be systematic variation in kangaroo posture with speed and mass which is yet to be fully explored.

In this study, we investigated the hindlimb kinematics and kinetics in kangaroos hopping at various speeds. Specifically, we explored the relationship between changes in posture, EMA, joint work, and tendon stress across a range of hopping speeds and body masses. To do this, we built a musculoskeletal model of a kangaroo based on empirical imaging and dissection data (*Figure 1a*). We used the musculoskeletal model to calculate ankle EMA throughout the stride by capturing changes in both muscle moment arm and the GRF moment arm, allowing us to explore their individual contributions. We hypothesised that (i) the hindlimb would be more crouched at faster speeds, primarily due to the distal hindlimb joints (ankle and metatarsophalangeal), independent of changes with body mass, and (ii) changes in moment arms resulting from the change in posture would contribute to the increase in tendon stress with speed, and may thereby contribute to energetic savings by increasing the amount of positive and negative work done by the ankle without requiring additional muscle work.

## Results

### Stride parameters

Hopping speed ranged from 1.99 to 4.48 ms$^{-1}$. Larger kangaroos tended to hop at slightly faster speeds ($B$=0.048, SE = 0.018, $p$=0.009, R$^2$=0.057), and due to this weak relationship between body mass and speed, both variables were considered in multiple linear regression models to determine their relative effects on the outcome measures (see *Appendix 1—table 1*).

Faster speeds were associated with a greater magnitude of acceleration in the braking period of the stance phase, i.e. minimum horizontal acceleration and maximum vertical acceleration (*Figure 2—figure supplement 1a*). There was no significant relationship between body mass and acceleration; however, there was a significant interaction between body mass and speed on maximum vertical acceleration, whereby smaller kangaroos had a greater change in vertical acceleration between slower and faster hopping speeds than larger kangaroos (*Appendix 1—table 1*).

Body mass and speed had different effects on ground contact duration (*Appendix 1—table 1*). There was a slightly greater (or longer) contact duration in larger kangaroos. A stronger, opposing relationship was found with speed, and as speed increased, contact duration decreased. The relatively tight correlation between contact duration and hopping speed (R$^2$=0.73) could prove useful for predicting speed when contact duration can be accurately measured. In *Figure 2—figure supplement 1b*, we combined our data with red kangaroo data from *Kram and Dawson, 1998* to extend the predictive range of both studies.

Larger kangaroos hopped with longer strides and lower frequencies than smaller kangaroos (*Appendix 1—table 1*), and stride length also increased with speed (*Figure 2—figure supplement 1c*). We found a significant decrease in stride frequency with mass and an increase with speed, if we did not consider the interaction term (*Figure 2—figure supplement 1d*). However, a significant interaction between body mass and speed suggests that larger kangaroos relied more on increases in stride frequency to increase hopping speed compared to smaller kangaroos (*Appendix 1—table 1*).

### Ground reaction forces

The stance begins with a braking phase (negative fore-aft horizontal GRF component), followed by a propulsive phase (positive fore-aft horizontal GRF component). At the transition from negative to positive, naturally, the GRF was vertical, and this occurred at 42.9±26.9% of stance. The peak in vertical GRF, however, occurred at 46.4±5.3% of stance (*Figure 2a and b*).

Larger kangaroos and faster speeds were associated with greater magnitudes of peak GRFs (*Figure 2a and b*; *Appendix 1—table 2*). When peak vertical GRF was normalised to body weight, peak forces ranged from 1.4 multiples of body weight (BW) to 3.7 BW per leg (mean: 2.41±0.396 BW) and increased with speed (*Figure 2—figure supplement 2a*). Smaller kangaroos may experience disproportionately greater GRFs at faster speeds (*Figure 2—figure supplement 2b*, *Appendix 1—table 2*). The increase in GRF will tend to increase tendon stress, irrespective of changes in posture.

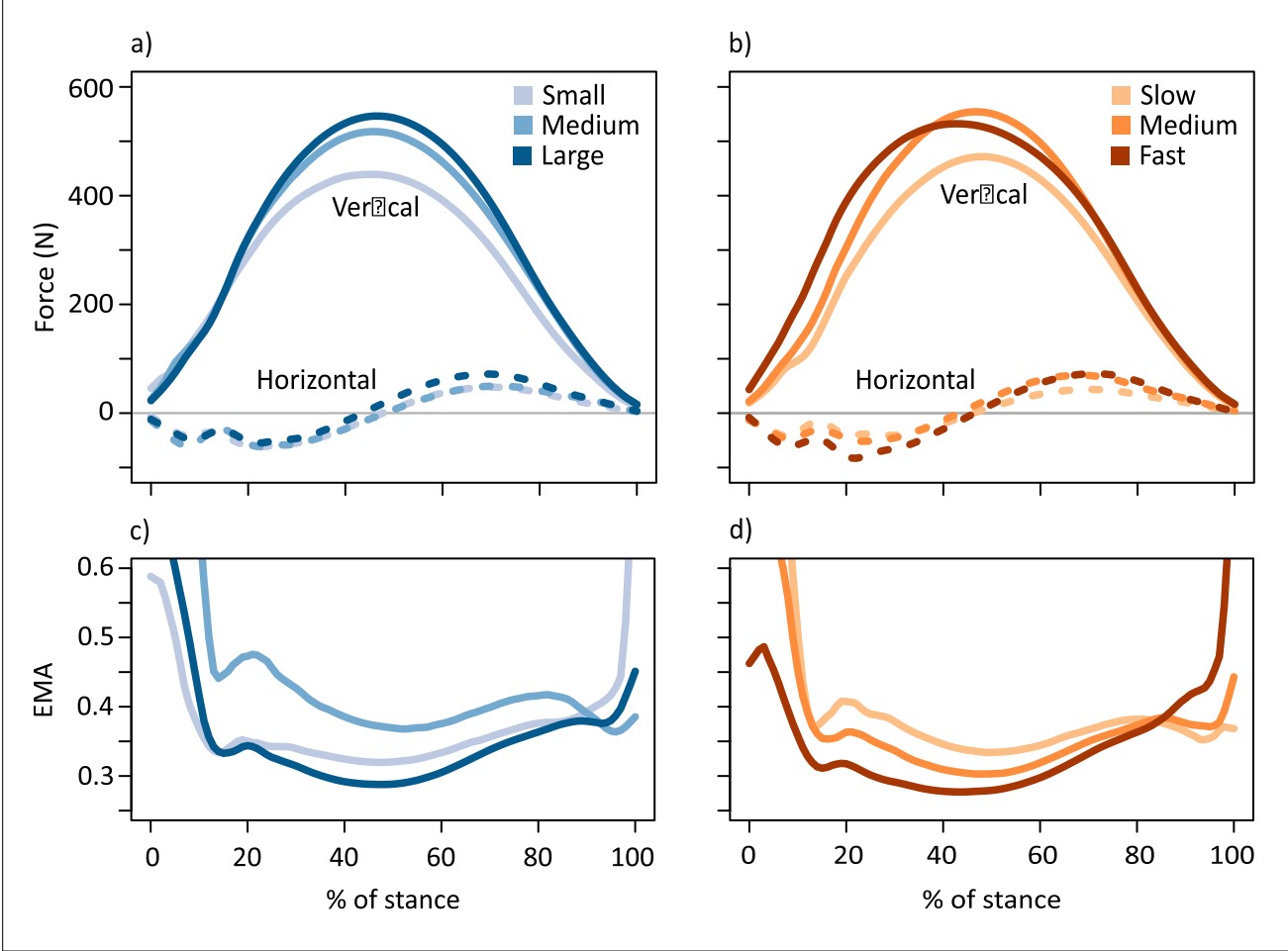

**Figure 2.** Horizontal fore-aft (dashed lines) and vertical (solid lines) components of the ground reaction force (GRF) (**a**) coloured by body mass subsets (small 17.6±2.96 kg, medium 21.5±0.74 kg, large 24.0±1.46 kg) and (**b**) coloured by speed subsets (slow 2.52±0.25 ms$^{-1}$, medium 3.11±0.16 ms$^{-1}$, fast 3.79±0.27 ms$^{-1}$). In (**a**) and (**b**), the medial-lateral component of the GRF is not shown as it remained close to zero, as expected for animals moving in a straight-line path. Lower panels show average time-varying effective mechanical advantage (EMA) for the ankle joint subset by (**c**) body mass and (**d**) speed.

The online version of this article includes the following figure supplement(s) for figure 2:

**Figure supplement 1.** Kinematics of the stride with speed and stance.

**Figure supplement 2.** Vertical ground reaction force with mass and speed.

**Figure supplement 3.** Peak vertical ground reaction force (GRF) plotted against tendon stress (*B*=0.080, SE = 0.009, *p*<0.001, R$^2$=0.486).

## Kinematics and posture

Kangaroo hindlimb stance phase kinematics varied with both body mass and speed. In partial support of hypothesis (i), greater masses and faster speeds were associated with more crouched hindlimb postures (*Figure 3a and c*; where crouch factor is the ratio of total limb length to pelvis to toe distance). Size-related changes in posture occurred throughout stance and were distributed as small changes among all hindlimb joints rather than a large shift in any one joint (*Figure 3d and f*; *Appendix 1—table 3*). The hip, knee, and ankle tended toward more flexion, and the metatarsophalangeal (MTP) toward greater extension (plantarflexion) in larger kangaroos.

In addition to, and independent of, the change in posture with mass, there were substantial postural changes due to speed (*Appendix 1—table 3*). Unlike with mass, speed-related changes occurred predominantly during the braking phase, and the changes were concentrated in the ankle and MTP joints, with little to no change in the proximal joints (*Figure 3c, e and g*; *Appendix 1—table 3*). There was a significant decrease in ankle plantarflexion at initial ground contact and increase in dorsi-flexion at midstance at faster speeds (*Figure 3g*; *Appendix 1—table 3*). Maximum ankle dorsiflexion

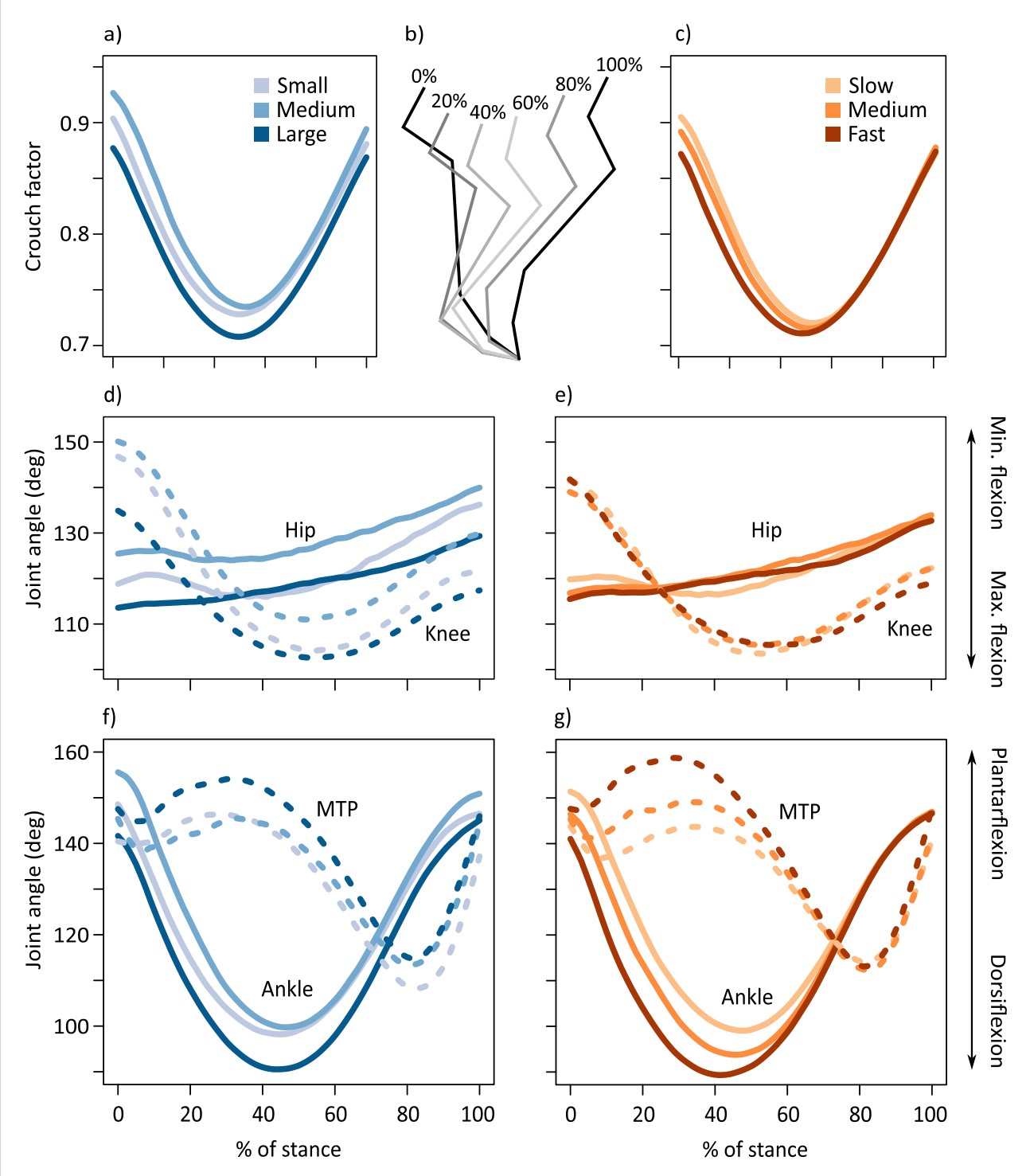

**Figure 3.** Average time-varying crouch factor (see *Figure 1a*) of the kangaroo hindlimb grouped by (**a**) body mass and (**c**) speed. Position of the limb segments during % stance intervals (**b**). Average time-varying joint angles for the hip (solid lines) and knee (dashed lines) displayed for kangaroos grouped by (**d**) body mass and (**e**) speed. Average time-varying joint angles for the ankle (solid lines) and metatarsophalangeal (MTP) joints (dashed lines) displayed for kangaroos grouped by (**f**) body mass and (**g**) speed. For (**f–g**), increased plantarflexion represents a decrease in joint flexion, while increased dorsiflexion represents increased flexion of the joint. Body mass subsets: small 17.6±2.96 kg, medium 21.5±0.74 kg, large 24.0±1.46 kg. Speed subsets: slow 2.52±0.25 ms⁻¹, medium 3.11±0.16 ms⁻¹, fast 3.79±0.27 ms⁻¹.

The online version of this article includes the following figure supplement(s) for figure 3:

*Figure 3 continued on next page*

*Figure 3 continued*

**Figure supplement 1.** Average time-varying gastrocnemius and plantaris muscle moment arm, *r*, grouped by (**a**) body mass and (**b**) speed; external moment arm to the ankle, *R*, grouped by (**c**) body mass, and (**d**) speed.

**Figure supplement 2.** Average time-varying net joint moments (dimensionless, as moments were divided by body weight * leg length) for the hip (solid lines) and knee (dotted lines) displayed for kangaroos grouped by (**a**) body mass and (**b**) speed.

occurred at 44.8±4.5% of stance, and tended to occur 3.9±0.7% earlier in stance with each 1 ms$^{-1}$ increase in speed ($p<0.001$). MTP range of motion increased with speed due to an increase in MTP plantarflexion prior to midstance rather than a change in dorsiflexion (*Figure 3g*).

Kangaroos were maximally crouched at midstance, with crouch factor (CF) reaching a minimum at 50.1±4.2% of stance. Crouch factor (CF) at initial ground contact decreased at faster speeds, although the limb was similarly flexed during midstance ($p=0.295$). Consequently, CF changed less at faster speeds than at slower speeds.

Larger kangaroos had smaller hip and knee ranges of motion (ROM) compared to smaller kangaroos (*Figure 3d*; *Appendix 1—table 3*). In the distal hindlimb, ankle ROM increased with body mass, largely owing to an increase in dorsiflexion at midstance (*Figure 3f*; *Appendix 1—table 3*). The ROM of the MTP joint did not change with body mass; however, there was both an increase in plantarflexion and a decrease in dorsiflexion, resulting in a shift to larger MTP angles with mass (*Figure 3f*; *Appendix 1—table 3*).

## Effective mechanical advantage

EMA of the ankle decreased as the limb became more crouched. The change in EMA with mass and speed was substantial, particularly in the braking period (*Figure 2c and d*). We evaluated the change in EMA at midstance, as it was approximately the point in the stride where GRF (and therefore tendon stress) was greatest. EMA at midstance decreased with body mass, although we did not detect an effect of speed (*Figure 2c and d*; *Appendix 1—table 5*), which may be due to an insignificant or undetected effect of speed on the external moment arm to the ankle, *R*. However, examining whether the decrease in EMA was caused by a decrease in the gastrocnemius and plantaris muscle moment arm, *r*, or an increase in *R* revealed a more nuanced picture.

Speed, rather than body mass, was associated with a decrease in *r* at midstance (*Figure 3—figure supplement 1a, b and h*; *Appendix 1—table 5*). A kangaroo travelling 1 ms$^{-1}$ faster would decrease *r* by approximately 4.2%. The muscle moment arm reduced with speed for the full range of speeds we measured (1.99–4.48 ms$^{-1}$), with the maximum *r* occurring when the ankle was at 114.4±0.8° (*Figure 3—figure supplement 1g*). Maximum ankle dorsiflexion ranged from 114.5° in the slowest trials, to 75.8° in faster trials (*Figure 3g*). The timing of the local minimum *r* at midstance coincided with the timing of peak ankle dorsiflexion.

We observed an increase in *R* with body mass, but not speed, at midstance (*Figure 3—figure supplement 1c, d and h*; *Appendix 1—table 5*). We expected *R* to vary with both mass and speed because the increase in MTP plantarflexion prior to midstance reduced the distance between the ankle joint and the ground, which would increase *R* since the centre of pressure (CoP) at midstance remained in the same position (*Appendix 1—table 2*). Increasing body mass by 4.6 kg or speed by 1 ms$^{-1}$ resulted in a~20% reduction in ankle height (vertical distance from the ground) (*Figure 3—figure supplement 1e and f*). If we consider the nonsignificant relationship between *R* (and EMA) and speed to indicate that there is no change in *R*, then it conflicts with the ankle height and CoP result. Taking both into account, we think it is more likely that there is a small, but important change in *R*, rather than no change in *R* with speed. It may be undetectable because we expect small effect sizes compared to the measurement range and measurement error (*Figure 3—figure supplement 1h*), or be obscured by a similar change in *R* with body mass. *R* is highly dependent on the length of the metatarsal segment, which is longer in larger kangaroos (1 kg BM corresponded to ~1% longer segment, $p<0.001$, R$^2$=0.449). If *R* does indeed increase with speed, then the changes in both *R* and *r* together will tend to decrease EMA at faster speeds.

## Joint moments

Despite the small changes in joint rotation in the proximal hindlimb, we detected a decrease in magnitude of the dimensionless hip extensor and knee flexor moments with mass (*Figure 3—figure*

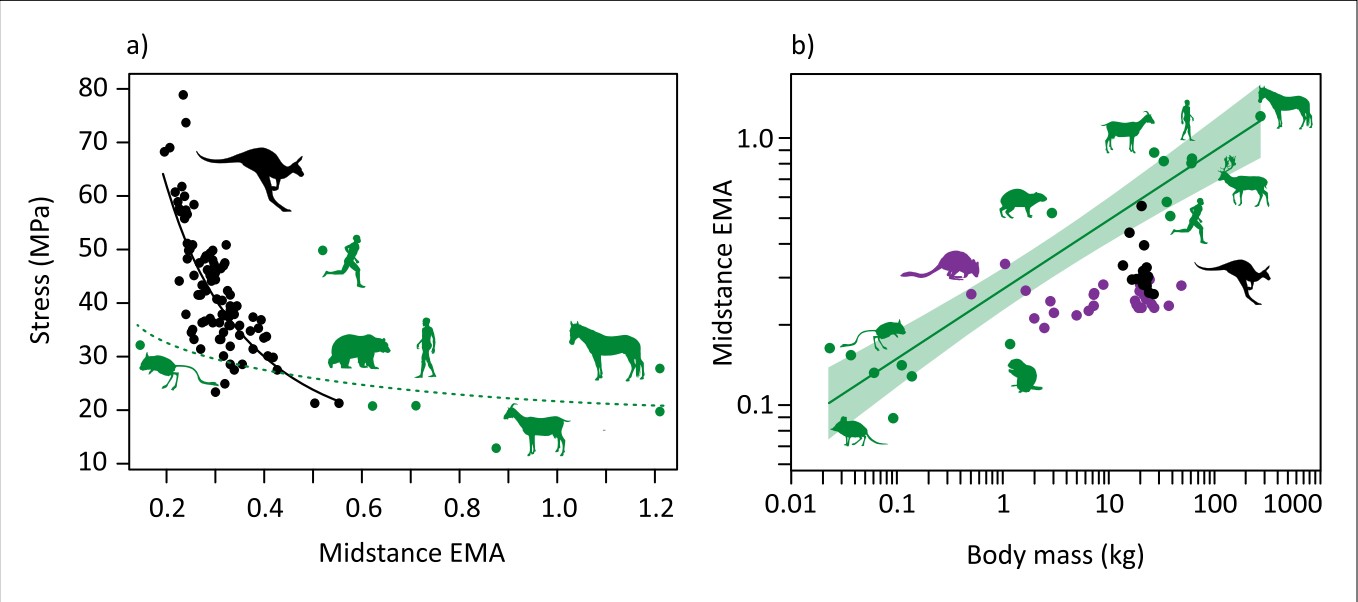

**Figure 4.** EMA with stress and mass. (**a**) Relationship between ankle effective mechanical advantage, EMA, at midstance and Achilles tendon stress (stress = 11.6 EMA$^{-1.04}$, $R^2$=0.593) (black), with other mammals (green). (**b**) Scaling of mean ankle EMA at midstance for each individual kangaroo against body mass (black), with data for a wider range of macropods (purple) (*Bennett and Taylor, 1995*), and other mammals (green, EMA = 0.269 $M^{0.259}$, shaded area 95% confidence interval) (*Biewener, 1990*) shown.

supplement 2a, *Appendix 1—table 4*), and an increase in magnitude of both joint moments with speed during the braking period of stance (*Figure 3—figure supplement 2b*). Maximum hip extensor moment and knee flexor moment were significantly influenced by the interaction between body mass and speed, suggesting that larger kangaroos increased the magnitude of the moments at a faster rate with speed compared to smaller kangaroos.

The change in the Achilles muscle moment arm, $r$, with speed has further implications for Achilles tendon stress, because force in the tendon is determined by the ratio of the ankle moment to $r$. As such, the decrease in $r$ with speed would tend to increase tendon force (and thereby tendon stress), as would an increase in ankle moment. We found that speed, rather than mass, was associated with an increase in the maximum ankle moment (*Figure 3—figure supplement 2*; *Appendix 1—table 4*). The range of body masses studied here may not be sufficient to detect an effect of mass on ankle moment in addition to the effect of speed.

## Tendon stress

Peak Achilles tendon stress increased as minimum EMA decreased (*Figure 4a*). Subtle decreases in EMA which may have been undetected in previous studies, correspond to discernible increases in tendon stress. For instance, reducing EMA from 0.242 (mean minimum EMA of the slow group) to 0.206 (mean minimum EMA of the fast group) was associated with an increase in tendon stress from ~50 MPa to ~60 MPa, decreasing safety factor from 2 to 1.67 (where 1 indicates failure), which is both measurable and physiologically significant. Tendon stress increased with body mass and speed (*Appendix 1—table 5*). Maximum stress in the ankle extensor tendons occurred at 46.8±4.9% of stance, aligning with the timing of peak vertical GRF and maximum ankle moment. Body mass did not have a significant effect on the timing of peak stress, but peak stress occurred 3.2±0.8% earlier in stance with each 1 ms$^{-1}$ increase in speed ($p$<0.001), reflecting the timing of the peak in ankle dorsi-flexion. The increase in tendon stress with speed, facilitated in part by the change in moment arms via the shift in posture, may explain changes in ankle work (c.f. Hypothesis (ii)).

## Joint work and power

An analysis of joint-level energetics showed the majority of the work per hop and power in the hind-limb was performed by the ankle joint (*Figure 5*; *Appendix 1—table 8*). The ankle performed nega-tive work in the braking period, as kinetic and gravitational potential energy was converted to elastic

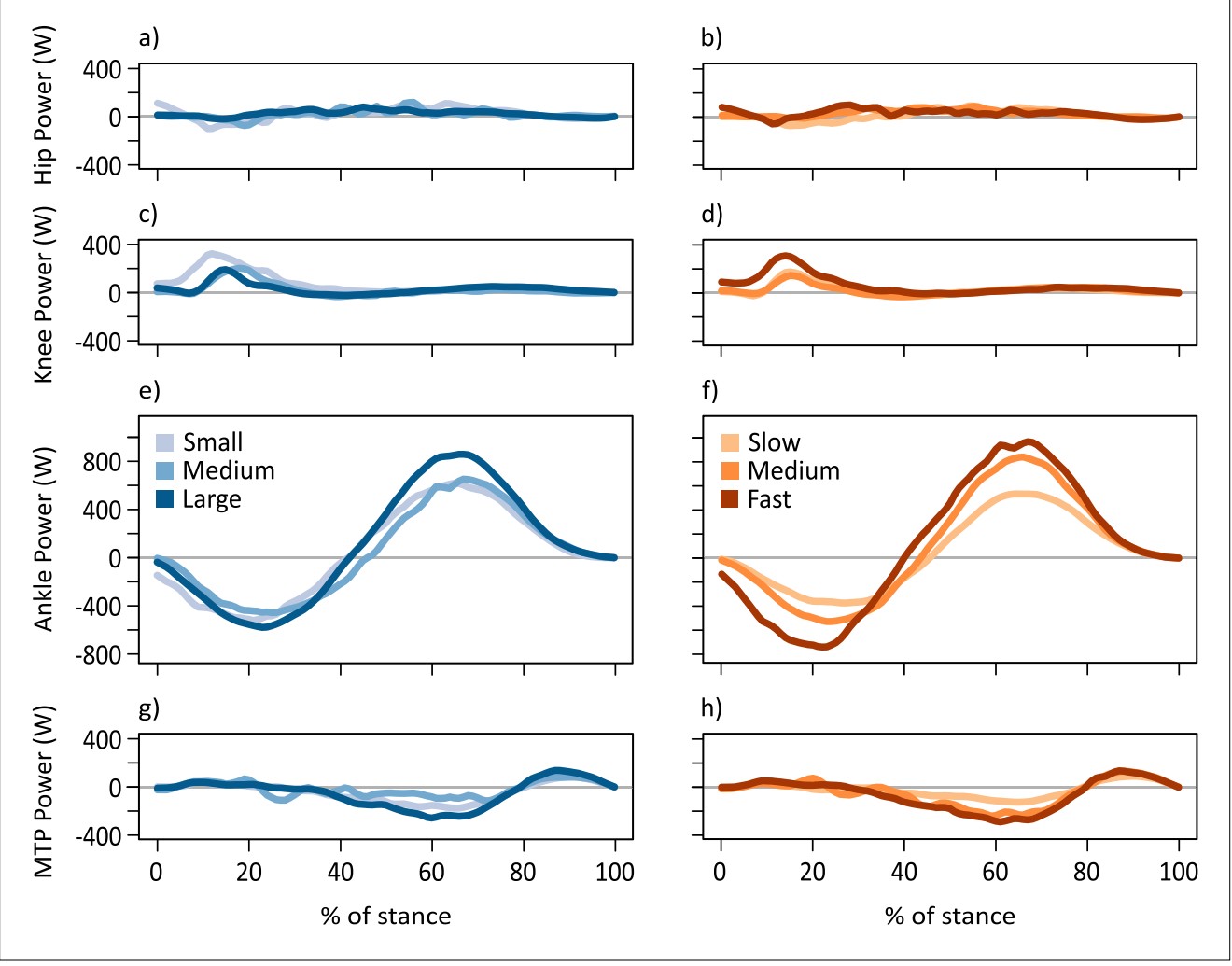

**Figure 5.** Time-varying joint powers. Average time-varying joint powers for the hip (**a, b**), knee (**c, d**), ankle (**e, f**), and MTP (**g, h**) displayed for kangaroos grouped by body mass (**a, c, e, g**) and speed (**b, d, f, h**). Power in all panels is set to the same scale. Body mass subsets: small 17.6±2.96 kg, medium 21.5±0.74 kg, large 24.0±1.46 kg. Speed subsets: slow 2.52±0.25 ms⁻¹, medium 3.11±0.16 ms⁻¹, fast 3.79±0.27 ms⁻¹.

potential energy by loading the ankle extensor tendons. An increase in negative ankle work was associated with the increase in tendon stress in the ankle extensors ($\beta$=0.750, SE = 0.074, $p$<0.001, $R^2$=0.511). Simple linear regression indicated that the increase in negative work was associated with speed but not body mass (*Figure 6a*; *Figure 6—figure supplement 1e and f*; *Appendix 1—table 6*).

Elastic potential energy was returned to do positive work in the propulsive period as the ankle started extending prior to midstance. Positive ankle work increased with both mass and speed (*Figure 6a*; *Figure 6—figure supplement 1e and f*; *Appendix 1—table 6*), but since it increased at the same rate as negative work, critically, there was no change in net ankle work with speed (*Figure 6b*; *Figure 6—figure supplement 2e and f*; *Appendix 1—table 6*). Thus, in support of hypothesis (ii), the increase in positive and negative ankle work may be due to the increase in tendon stress rather than additional muscle work.

The MTP was the only joint which did predominantly negative work. The MTP did more negative work with speed, while net MTP work decreased with both mass and speed (*Figure 6—figure supplement 1g and h*; *Figure 6—figure supplement 2g and h*; *Appendix 1—table 6*). The MTP transition from negative work to positive work at ~80% stance, as the back of the foot started to leave the ground. Conversely, the knee did almost no negative work, and the hip did very little (*Figure 5*; *Figure 6—figure supplements 1 and 2*). Net work slightly increased with mass and speed in the hip, while at the knee, net work increased only with speed (*Appendix 1—table 6*).

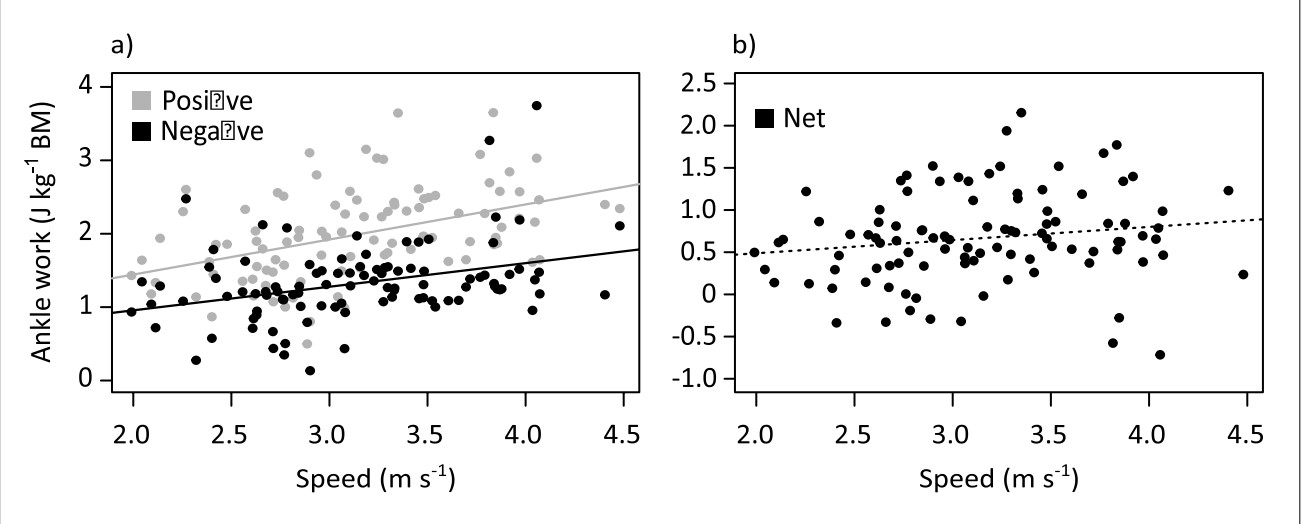

**Figure 6.** Variation with speed of (**a**) positive and negative ankle work, and (**b**) net ankle work per hop.

The online version of this article includes the following figure supplement(s) for figure 6:

**Figure supplement 1.** Positive (purple) and negative (green) joint work over stance for the hip, knee, ankle, and metatarsophalangeal (MTP) plotted against body mass (**a, c, e, g**) and speed (**b, d, f, h**).

**Figure supplement 2.** Net joint work for the hip, knee, ankle, and metatarsophalangeal (MTP) joint over stance plotted against body mass (**a, c, e, g**) and speed (**b, d, f, h**).

**Figure supplement 3.** Ankle work with EMA at midstance.

## Discussion

### How does posture contribute to kangaroo energetics?

The cost of generating force hypothesis (*Taylor et al., 1980*) implies that as animals increase loco-motor speed and decrease ground contact time, metabolic rate should increase (*Kram and Taylor, 1990*). Macropods defy this trend (*Dawson and Taylor, 1973*; *Baudinette et al., 1992*; *Kram and Dawson, 1998*). It is likely that the use of their ankle extensor tendons set them apart from other mammals, but the underlying mechanisms are unclear (*Bennett and Taylor, 1995*; *Bennett, 2000*; *Thornton et al., 2022*). We hypothesised, and found, that the hindlimbs were more crouched at faster speeds, primarily due to the ankle and metatarsophalangeal joints. We propose that changes in moment arms (EMA) resulting from the change in posture would contribute to the increase in tendon stress with speed, and may thereby contribute to energetic savings by increasing the amount of positive and negative work done by the ankle, without requiring additional muscle work (*Figure 7*).

Achilles tendon stress depends on extensor muscle force, which in turn depends on the muscle moment arm, ankle moment, external moment arm, and GRF, assuming tendon material properties and cross-sectional area remain unchanged (*Figure 7*). We detected an increase in peak ankle moment with speed independent of increases in body mass, despite the change in the external moment arm, $R$, with mass, likely due to a dominant effect of the GRF (ankle moment = GRF · $R$). Peak GRF also naturally increased with speed together with shorter ground contact durations (*Figures 1b and 2b*, *Figure 2—figure supplement 1*). However, both the increase in ankle moment and decrease in the internal moment arm, $r$, with speed tend to increase tendon stress (stress = tendon force/cross-sectional area; tendon force = ankle moment/$r$). Although the increase in peak GRF with speed explains much of the increase in tendon stress (and strain) (*Figure 2—figure supplement 3*), the significant relationship between tendon stress and ankle EMA suggests that the increase in stress is not solely due to the greater GRFs, but that stress is also modulated by changes in posture throughout the stride (*Figure 4a*). If peak GRF were constant but EMA changed from the average value of a slow hop to a fast hop, then stress would increase 18%, whereas if EMA remained constant and GRF varied by the same principles, then stress would only increase by 12%. Thus, changing posture and decreasing ground contact duration both appear to influence tendon stress for kangaroos, at least for the range of speeds we examined.

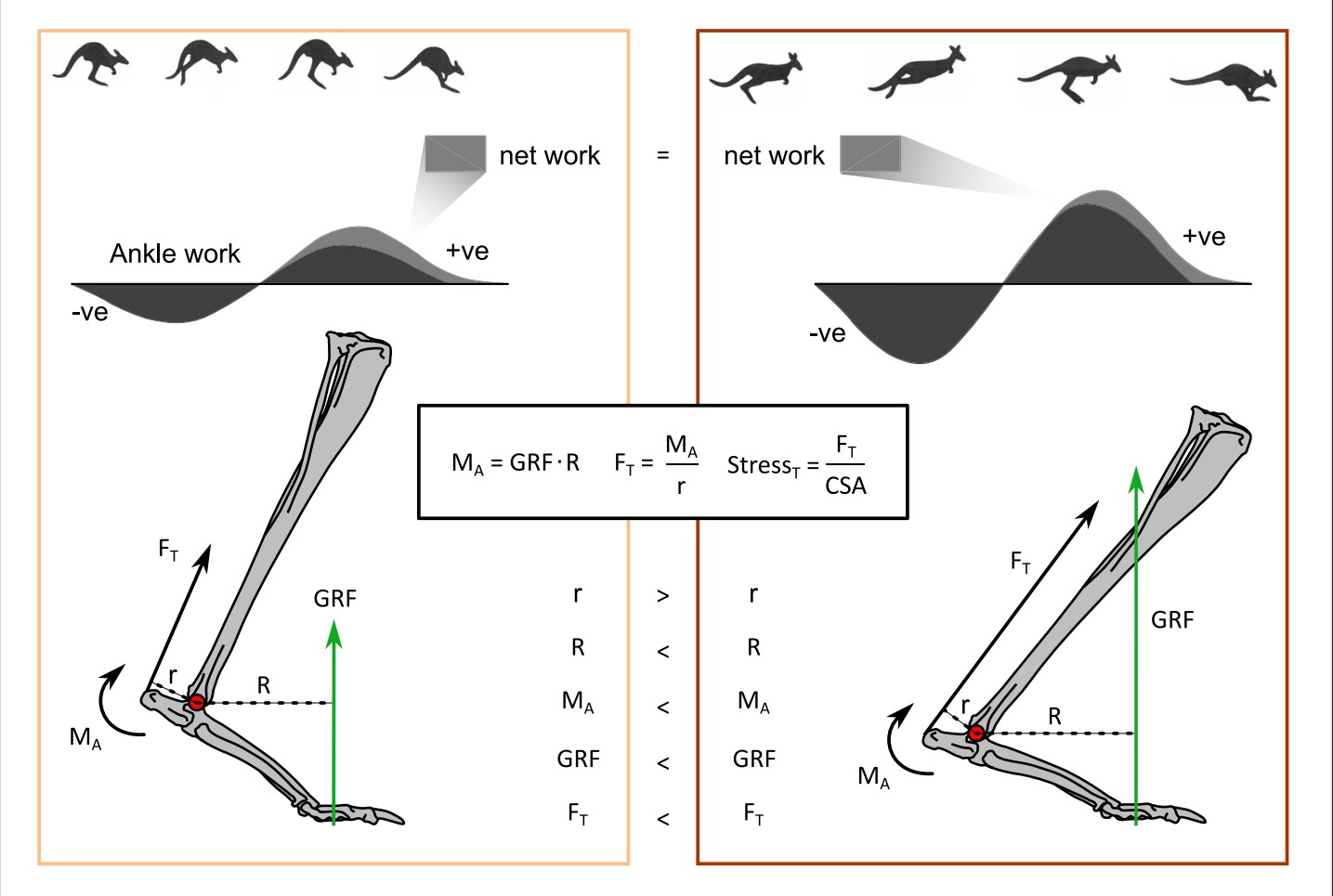

**Figure 7.** How the relationship between posture and speed is proposed to change tendon stress. Forces are not to scale and joint angles are exaggerated for illustrative clarity. A slow hop (left panel) compared to a fast hop (right panel). The increase in ground reaction force (GRF) with speed, while a more crouched posture changes the muscle moment arm, r, and external moment arm, R, which allows the ankle to do more negative work (storing elastic potential energy in the tendons due to greater tendon stresses), without increasing net work, and thereby metabolic cost. Ankle moment was calculated by OpenSim and includes inertial terms.

Previously, small and large macropods were reported to have consistent ankle EMA due to the muscle moment arm, r, and the external moment arm, R, scaling similarly with body mass (*Figure 4b*; *Bennett and Taylor, 1995*). Our results, however, suggest posture may not be as consistent as previously reported; we found that EMA and moment arms varied with both mass and speed. Larger and faster kangaroos were more crouched, leading to smaller ankle EMA. We were able to explore the individual contributions to ankle EMA using our musculoskeletal model to demonstrate that changes in the ankle and MTP range of motion resulted in changes in *r* with speed, and *R* with mass, respectively. *R* (and therefore also EMA) may also increase with speed, given that the ankle joint lowers substantially closer to the ground. These effects of the moment arms were additive; combined, there was a marked decrease in ankle EMA, particularly during the early stance when the ankle extensor tendons are loaded (*Figure 2c and d*).

Given that tendon stress seems to be partially controlled by EMA, kangaroos could change their posture to absorb and return greater amounts of elastic strain energy, to meet the requirements of increasing positive ankle work at faster speeds. A comparison among all the hindlimb joints suggests the ankle was primarily responsible for both the storage and release of energy during the stride (*Figure 5*). Positive work increased with mass and speed, consistent with other studies in other species (*Cavagna and Kaneko, 1977*). However, the amount of negative work absorbed during the stride also increased with speed, while, critically, net ankle work did not change, which indicates that the increase

in negative work that was absorbed matched the increase in positive work that was required to move forward at more rapid speeds (*Figure 6*). Indeed, we observed that all trials had a similar positive net ankle work per hop observed across all speeds (mean: 0.67±0.54 Jkg$^{-1}$). The consistent net work observed among all speeds (combined with the consistent stride frequency) suggests that the ankle extensor muscle-tendon units were performing similar amounts of ankle work per hop independent of speed, which would predominantly be done by the tendon. As such, the change in ankle EMA appears to be a mechanism that contributes to the increase in tendon stress and positive ankle work. This mechanism may help explain why hopping macropods do not follow the energetic trends observed in other species (*Dawson and Taylor, 1973*; *Baudinette et al., 1992*; *Kram and Dawson, 1998*), but it does not fully resolve the cost of generating force conundrum.

## Decoupling force and energy during hopping

As Achilles tendon stress increases with speed in hopping kangaroos, so too must the muscle force in the gastrocnemius and plantaris; otherwise, the tendon would lengthen the muscle rather than store elastic energy. These increases in force could be achieved via increases in muscle activation, but this would also increase muscle-level energy consumption, and thus metabolic energy use. The question remains; whether there are physiologically plausible scenarios that could lead to an increase in muscle force without increasing energy use. Experimental data from hopping macropods confirm that tendon force and EMG can be decoupled, whereby greater EMG does not translate to larger forces or tendon stresses (*Biewener et al., 2004b* Table 2). Thus, the decoupling of energetics and speed might be explained by a muscle-level change in mechanics, driven by either force-length or force-velocity relationships. Force-length effects might explain this result if muscles could operate at more favourable regions of their force-length relationship during tendon loading. The gastrocnemius and plantaris muscles of tammar wallabies do not appear to shorten considerably during hopping, but rather are suggested to act near-isometrically as a strut; however,, this near-isometric behaviour only occurs in the second half of the stance phase (*Biewener et al., 1998*). This pattern in muscle fibre length does not vary with hopping speed, despite large increases in muscle tendon forces (*Biewener et al., 1998*).

Force-velocity effects may instead play an important role. During hopping, the gastrocnemius and plantaris activate prior to ground contact, and the majority of force development during the first half of stance is associated with muscle fibre stretch, such that the fibres undergo substantial active lengthening prior to their near-isometric behaviour (*Biewener et al., 2004b*, *Figure 3*). Faster hopping speeds are also accompanied by decreased contact durations, suggesting that the rate of this muscle stretch likely increases with speed. This is supported with a tendon force transducer (buckle) and kinematic data of kangaroos hopping at faster speeds (*Griffiths, 1989*). Griffiths demonstrated that faster speeds lead to greater muscle stretch rates, the result of which enables rates of force to rise 10 times faster when compared to supramaximal isometric contractions in the same muscle. Experiments exploring single fibre energetics in frogs show that active muscle lengthening is less costly compared to isometric (*Linari et al., 2003*), allowing the muscle to exert tension economically. Thus, the changes in posture associated with increases in speed may allow increases in active muscle fibre stretch rates during the first half of stance, supporting increases in tendon stress, and thereby also energy storage, without increases in activation or metabolic cost. However, a detailed musculoskeletal model that incorporates individual muscle mechanical and energetic models is necessary to confirm this phenomenon.

## Considerations and limitations

First, we believe it is more likely that the changes in moment arms and EMA can be attributed to speed rather than body mass, given the marked changes in joint angles and ankle height observed at faster hopping speeds. However, our sample included a relatively narrow range of body masses (13.7–26.6 kg) compared to the potential range (up to 80 kg), limiting our ability to entirely isolate the effects of speed from those of mass. Future research should examine a broader range of body sizes. Second, kangaroos studied here only hopped at relatively slow speeds, which bounds our estimates of EMA and tendon stress to a less critical region. As such, we were unable to assess tendon stress at fast speeds, where increased forces would reduce tendon safety factors closer to failure. A different experimental or modelling approach may be needed, as kangaroos in enclosures seem unwilling to hop faster over force plates. Finally, we did not determine whether the EMA of proximal hindlimb

joints (which are more difficult to track via surface motion capture markers) remained constant with speed. Although the hip and knee contribute substantially less work than the ankle joint (*Figure 5*), the majority of kangaroo skeletal muscle is located around these proximal joints. A change in EMA at the hip or knee could influence a larger muscle mass than at the ankle, potentially counteracting or enhancing energy savings in the ankle extensor muscle-tendon units. Further research is needed to understand how posture and muscles throughout the whole body contribute to kangaroo energetics.

## Why are macropods unique?

No other mammals are known to achieve the same energetic feat as macropods, despite similar tendons or stride parameters (*Thornton et al., 2022*), but macropods are unique in other ways. Kangaroos operate at much lower ankle EMA values than other large mammalian species (*Figure 4b*). Mammals >18 kg tend to operate with EMA values of 0.5–1.2. The only mammals with comparable EMA values to kangaroos are rodents <1 kg (*Figure 4b*; *Biewener, 1990*), such as kangaroo rats which have relatively thicker tendons that may be less suited for recovering elastic strain energy (*Biewener and Blickhan, 1988*) (but see: *Christensen et al., 2022*). *Figure 4a* shows the non-linear relationship between tendon stress and EMA in kangaroos, quadrupeds, and humans. The range of EMA estimates for other mammals suggests they operate in the region of this curve where large changes in EMA would only produce small changes in stress. This implies EMA modulation in other species is not as effective a mechanism to increase tendon stress with increased running speed as that observed in kangaroos.

## Were macropods performance- or size-limited?

The morphology and hopping gait that make kangaroos supremely adapted for economical locomotion likely also have several performance limitations. One possible consequence is a predicted reduction in manoeuvrability (*Biewener, 2005*). The high compliance of the Achilles tendon would limit the ability to rapidly accelerate, owing to the time lag between muscle force production and the transmission of this force to the environment. A second important limitation is the requirement for kangaroos to operate at large tendon stresses. Previous research has suggested that kangaroos locomote at dangerously low safety factors for tendon stress, predicted to be between 1–2 for large kangaroos (*Kram and Dawson, 1998*; *McGowan et al., 2008*; *Snelling et al., 2017*; *Thornton et al., 2022*). Our new insights into the mass and speed modulated changes in EMA suggest that these safety factors may be even lower, likely limiting the maximum body mass that hopping kangaroos can achieve. This suggests that previous projections of tendon stress may have overestimated the body mass at which tendons reach their safety limit ('safety factor' of 1). *Snelling et al., 2017* and *McGowan et al., 2008* estimated the maximum body mass to remain above this limit was approximately 150 kg, but even if we consider this is a conservative prediction, it is far lower than the estimated mass of extinct macropodids (up to 240 kg *Helgen et al., 2006*; *Janis et al., 2023*). Thus, we expect there must be a body mass where postural and gait changes shift from contributing to stress to mitigating it (*Dick and Clemente, 2017*).

This study highlights how EMA may be more adjustable than previously assumed, and how musculoskeletal modelling and simulation approaches can provide insights into direct links between form and function which are often challenging to determine from experiments alone.

## Methods
### Animals and data collection

Hopping data for red and eastern grey kangaroos was collected at Brisbane's Alma Park Zoo in Queensland, Australia, in accordance with approval by the Ethics and Welfare Committee of the Royal Veterinary College; approval number URN 2010 1051. The dataset includes 16 male and female kangaroos ranging in body mass from 13.7 to 26.6 kg (20.9±3.4 kg). Two juvenile red (*Macropus rufus*) and 11 grey kangaroos (*Macropus giganteus*) were identified, while the remaining three could not be differentiated as either species. Body mass was determined in several ways, primarily by measurements of the kangaroos standing stationary on the force plate. If the individual did not stop on the force plate, and forward velocity was constant, then body mass was determined by dividing the total vertical impulse (force integrated across time) across a constant velocity hop cycle (foot strike to foot

strike) by the total hop cycle time and further dividing by gravitational acceleration. Finally, if neither of these approaches were sufficient, we interpolated from the relationship between leg marker distances (as proxy for segment lengths) and body mass. These methods produce estimates of body masses close to stationary measurements, where both were available.

## Experimental protocol

Kangaroos hopped down a runway (~10×1.5 m) that was constructed in their enclosure using hessian cloths and stakes (*Figure 1—video 1*). The kangaroos elected to hop between 1.99 and 4.48 ms⁻¹, with a range of speeds and number of trials for each individual (*Figure 1—figure supplement 1*). The runway was open at both ends with two force plates (Kistler custom plate (60×60 cm) and AMTI Accugait plate (50×50 cm)) set sequentially in the centre and buried flush with the surface. The force plates recorded ground reaction forces (GRF) in the vertical, horizontal, and lateral directions.

A 6-camera 3D motion capture system (Vicon T160 cameras), recorded by Nexus software (Vicon, Oxford, UK) at 200 Hz, was used to record kinematic data. Reflective markers were placed on the animals over the estimated hip, knee, ankle, MTP joints; the distal end of phalanx IV; the anterior tip of the ilium; and the base of the tail (*Figure 1a*). Force plate data was synchronously recorded at 1000 Hz via an analogue-to-digital board integrated with the Vicon system.

## Data analysis

We calculated the stride length from the distance between the ankle or MTP marker coordinates at equivalent time points in the stride. The stance phase was defined as the period when the vertical GRF was greater than 2% of the peak GRF. We determined ground contact duration and total stride duration from the frame rate (200 Hz) and the number of frames from contact to take-off (contact duration) and contact to contact (stride duration), respectively, and calculated the stride frequency.

In most trials, the stride before and after striking the force plate was visible, providing a total of 173 strides. If two strides were present in a trial, we took the average of the two strides for that trial. Trials were excluded if only one foot landed on the force plate or if the feet did not land near-simultaneously, to give a total of 100 trials, with variety in the number of trials per kangaroo (*Figure 1—figure supplement 1*). We did not include trials where the kangaroo started from or stopped on the force plate. We assumed GRF was equally shared between each leg and divided the vertical, horizontal, and lateral forces in half, and calculated all results for one leg. We normalised GRF by body weight.

The fore-aft movement of the centre of pressure (CoP) was recorded by the force plate within the motion capture coordinate system (*Figure 1b*). We assumed that the force was applied along phalanx IV and that there was no medial-lateral movement of the CoP. It was necessary to assume the CoP was fixed along the medial-lateral axis because when two feet landed on the force plate, the lateral forces on each foot were not recorded, and indeed should have cancelled if the forces were symmetrical (i.e. if the kangaroo was hopping in a straight path and one foot is not in front of the other). We only used symmetrical trials to ensure reliable measures of the anterior-posterior movement of the CoP.

We calculated the hopping velocity and acceleration of the kangaroo in each trial from the position of the pelvis marker, as this marker was close to the centre of mass and there should have been minimal skin movement artefact. Position data were smoothed using the 'smooth.spline' function in R (version 3.6.3). The average locomotor speed of the trial was taken as the mean horizontal component of the velocity during the aerial phase before and after the stance phase.

## Building a kangaroo musculoskeletal model

We created a musculoskeletal model based on the morphology of a kangaroo for use in OpenSim (v3.3; *Seth et al., 2018*; *Figure 1a*). The skeletal geometry was determined from a computed tomography (CT) scan of a mature western grey kangaroo (https://www.morphosource.org/, Duke University, NC, USA). Western grey kangaroos are morphologically similar to eastern grey and red kangaroos (*Thornton et al., 2022*).

We extracted the skeletal components from the CT scan using Dragonfly (Version 2020.2, Object Research Systems (ORS) Inc, Montreal, Canada) and partitioned the hindlimb into five segments (pelvis, femur, tibia, metatarsals and calcaneus, and phalanges). The segments were imported into Blender (version 3.0.0, https://www.blender.org/; Amsterdam, Netherlands) to clean and smooth the bones, align the vertebrae, and export the segments as meshes. We imported the meshes into

Rhinoceros (version 6.0, Robert McNeel & Associates, Seattle, WA, USA) to construct the framework of the movement system.

All joints between segments were modelled as hinge joints which constrain motion to a single plane and one degree of freedom (DOF), except the hips which were modelled as ball-and-socket joints with three DOF. The joints were restricted to rotational (no translational) movement. The joints were marked with an origin (joint centre) and a coordinate system determined by the movement of the segment (x is abduction/adduction, y is pronation/supination, z is extension/flexion). The limb bones and joints from one leg were mirrored about the sagittal plane to ensure bilateral symmetry.

The segment masses for a base model were determined from measurements of eastern grey and red kangaroos provided in *Hopwood, 1976*. Cylinders set to the length and mass of each segment were used to approximate the segment centre of mass and moments of inertia.

We scaled the model to the size, shape, and mass of each kangaroo using the OpenSim scale tool and kangaroo markers. A static posture was defined based on the 3D positions of the markers at midstance. Each segment was scaled separately, allowing for different scaling factors across segments but preserving mass distribution. The markers with less movement of the skin over the joint (e.g. the MTP marker) were more highly weighted than the markers with substantial skin movement (e.g. the hip and knee markers) in the scaling tool.

## Joint kinematics and mechanics

We used inverse kinematics to determine time-varying joint angles during hopping. Inverse kinematics is an optimisation routine which simulates movement by aligning the model markers with the markers in the kinematic data for each time step (*Figure 1—video 2*). We adjusted the weighting on the markers based on the confidence and consistency in the marker position relative to the skeleton. The model movement from inverse kinematics was combined with GRFs in an inverse dynamics analysis in OpenSim to calculate net joint moments for the hip, knee, ankle, and MTP joints throughout the stance phase. Joint moments were normalised to body weight and leg length (*Figure 1a*).

We calculated instantaneous joint powers (joint work) for each of the hindlimb joints over the stance phase of hopping as the product of joint moment (not normalised) and joint angular velocity. To determine joint work, we integrated joint powers with respect to time over discrete periods of positive and negative work, consistent with *Dick et al., 2019*. For the stance duration of each hop, all periods of positive work were summed and all periods of negative work were summed to determine the positive work, negative work, and net work done at each of the hindlimb joints for a hop cycle. Joint work was normalised to body mass.

## Posture and EMA

We evaluated overall hindlimb posture to determine how crouched or upright the hindlimbs were during the stance phase of hopping. The total hindlimb length was determined as the sum of all segment lengths between the joint centres, from the toe to ilium (*Figure 1a*). The crouch factor (CF) was calculated as the distance between the toe and the ilium marker divided by the total hindlimb length. Larger CF values indicated extended limbs, whereas smaller CF values indicated more crouched postures.

We calculated the EMA at the ankle as the muscle moment arm of the combined gastrocnemius and plantaris tendon, *r*, divided by the external moment arm, *R* (perpendicular distance between the GRF vector and the ankle joint) (*Figure 1b*). We dissected (with approval from the University of the Sunshine Coast Ethics Committee, ANE2284) a road-killed 27.6 kg male eastern grey kangaroo and used this, combined with published anatomy on the origin and insertion sites, to determine the ankle extensor muscle-tendon unit paths on the skeleton in OpenSim (*Bauschulte, 1972*; *Hopwood and Butterfield, 1976*; *Hopwood and Butterfield, 1990*). The value *r* to the gastrocnemius and plantaris tendons for all possible ankle angles were determined from OpenSim and scaled to the size of each kangaroo, while *R* was calculated as the perpendicular distance between the ankle marker and the GRF vector at the CoP.

## Tendon stress

Ankle extensor tendon forces were estimated as the time-varying ankle moment divided by the time-varying Achilles tendon moment arm. Achilles tendon cross-sectional area was determined as the sum

of the gastrocnemius and plantaris tendon cross-sectional areas, which were determined for each kangaroo body mass based on area-mass regressions from *Snelling et al., 2017*. Forces were divided by tendon cross-sectional area to calculate tendon stress. We excluded the third ankle extensor tendon (flexor digitorum longus), which has a shorter muscle moment arm and which stores ~10% of strain energy of the ankle extensors in tammar wallabies, as it is primarily involved in foot placement rather than energy storage (*Biewener and Baudinette, 1995*).

## Statistics

We used multiple linear regression (lm function in R, v. 3.6.3, Vienna, Austria) to determine the effects of body mass and speed on the stride parameters, ground reaction forces, joint angles and CF, joint moments, joint work and power, and tendon stress. We considered the interaction of body mass and speed first, and removed the interaction term from the linear model if the interaction was not significant. Effects were considered significant at the $p < 0.05$ level. Each trial was treated individually, and species was not used as a factor in the analysis, as there was no systematic difference in outcome measures between kangaroo species. The data was grouped into body mass (small 17.6±2.96 kg, medium 21.5±0.74 kg, large 24.0±1.46 kg) and speed (slow 2.52±0.25 ms⁻¹, medium 3.11±0.16 ms⁻¹, fast 3.79±0.27 ms⁻¹) subsets for display purposes only.

## Acknowledgements

We thank the Brisbane Alma Park Zoo for hosting and facilitating this work, particularly Dena Loveday and Heather Hesterman. Matthew Brown provided access to the CT scan, the collection of which was funded by the Texas Vertebrate Paleontology Collections. Megan Johnston and Rachel Lyons from Wildcare Australia provided road-killed kangaroos for muscle dissection. We also thank Alex Muir from Logemas for loan of the 3D motion capture system. We also thank the reviewers for their suggestions and edits to the manuscript, their contribution greatly strengthened this manuscript. This work was supported by: Australian Government Research Training Program Scholarship and ISB Comparative Neuromuscular Biomechanics Technical Group Student Grant-in-Aid of Research to LHT; Biotechnology and Biological Sciences Research Council Grant (BB/F000863) to JRH; Journal of Experimental Biology Travelling Fellowship to CPM; Australian Research Council Discovery Project Grant to CJC, TD, and CM (DP230101886).

## Additional information

### Funding

| Funder | Grant reference number | Author |
| --- | --- | --- |
| Australian Research Council | DP230101886 | Taylor Dick<br>Christofer J Clemente |
| Biotechnology and Biological Sciences Research Council | BB/F000863 | John R Hutchinson |
| Comparative Neuromuscular Biomechanics Technical Group | Student Grant-in-Aid of Research | Lauren Thornton |
| Journal of Experimental Biology | Travelling Fellowship | Craig P McGowan |

The funders had no role in study design, data collection and interpretation, or the decision to submit the work for publication.

### Author contributions

Lauren Thornton, Data curation, Formal analysis, Investigation, Writing – original draft, Writing – review and editing; Taylor Dick, Data curation, Funding acquisition, Writing – original draft, Writing – review and editing; John R Hutchinson, Glen A Lichtwark, Craig P McGowan, Jonas Rubenson, Alexis

Wiktorowicz-Conroy, Data curation, Methodology; Christofer J Clemente, Conceptualization, Data curation, Supervision, Funding acquisition, Methodology, Writing – original draft, Project administration, Writing – review and editing

## Author ORCIDs
Lauren Thornton ⓘ https://orcid.org/0000-0002-9667-0951
Taylor Dick ⓘ https://orcid.org/0000-0002-7662-9716
John R Hutchinson ⓘ https://orcid.org/0000-0002-6767-7038
Glen A Lichtwark ⓘ https://orcid.org/0000-0001-7366-3348
Craig P McGowan ⓘ https://orcid.org/0000-0002-5424-2887
Jonas Rubenson ⓘ https://orcid.org/0000-0003-2854-1776
Christofer J Clemente ⓘ https://orcid.org/0000-0001-8174-3890

## Ethics
Hopping data for red and eastern grey kangaroos was collected at Brisbane's Alma Park Zoo in Queensland, Australia, in accordance with approval by the Ethics and Welfare Committee of the Royal Veterinary College; approval number URN 2010 1051.

Reviewer #1 (Public review): https://doi.org/10.7554/eLife.96437.3.sa1
Author response https://doi.org/10.7554/eLife.96437.3.sa2

---

# Additional files

## Supplementary files
MDAR checklist

## 1Data availability
Data and detailed analysis are available here https://doi.org/10.6084/m9.figshare.30282592.v1.

The following dataset was generated:

| Author(s) | Year | Dataset title | Dataset URL | Database and Identifier |
|---|---|---|---|---|
| Christofer C, Taylor D, Thornton LH | 2025 | Postural adaptations may contribute to the unique locomotor energetics seen in hopping kangaroos | https://doi.org/10.6084/m9.figshare.30282592.v1 | figshare, 10.6084/m9.figshare.30282592.v1 |
| | | | , | |

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

# Appendix 1

**Appendix 1—table 1.** Stride parameter multiple linear regression results as slopes, standard errors, and p-values.

Models with a significant interaction are displayed in full, and as a simplified model without the interaction term included (marked *). The fit of the model is represented by $R^2$ and relationships are considered significant at $p<0.05$.

| | Interaction | | | Body mass | | | Speed | | | $R^2$ |
|---|---|---|---|---|---|---|---|---|---|---|
| | β | SE | P | β | SE | P | β | SE | P | |
| Maximum vertical acceleration (ms⁻²) | −1.19 | 0.567 | 0.038 | 2.72 | 1.64 | 0.100 | 32.5 | 12.5 | 0.011 | 0.190 |
| Maximum vertical acceleration (ms⁻²)* | | | | −0.673 | 0.294 | 0.024 | 6.32 | 1.60 | 0.000 | 0.152 |
| Minimum vertical acceleration (ms⁻²) | | | | 0.221 | 0.208 | 0.290 | −0.268 | 1.13 | 0.814 | 0.012 |
| Maximum horizontal acceleration (ms⁻²) | | | | −0.036 | 0.296 | 0.903 | 0.159 | 1.61 | 0.922 | 0.000 |
| Minimum horizontal acceleration (ms⁻²) | | | | 0.407 | 0.370 | 0.275 | −6.05 | 2.02 | 0.003 | 0.087 |
| Contact duration (ms) | | | | 2.87 | 0.393 | 0.000 | −34.2 | 2.12 | 0.000 | 0.735 |
| Stride length (m) | | | | 0.014 | 0.004 | 0.001 | 0.376 | 0.020 | 0.000 | 0.815 |
| Stride frequency (Hz) | 0.022 | 0.008 | 0.007 | −0.082 | 0.023 | 0.001 | −0.392 | 0.180 | 0.032 | 0.303 |
| Stride frequency (Hz)* | | | | −0.019 | 0.004 | 0.000 | 0.102 | 0.023 | 0.000 | 0.245 |

**Appendix 1—table 2.** Ground reaction force and centre of pressure (CoP) multiple linear regression results as slopes, standard errors, and p-values.

Models with a significant interaction are displayed in full, and as a simplified model without the interaction term included (marked *). The fit of the model is represented by $R^2$ and relationships are considered significant at $p<0.05$.

| | Interaction | | | Body mass | | | Speed | | | $R^2$ |
|---|---|---|---|---|---|---|---|---|---|---|
| | β | SE | P | β | SE | P | β | SE | P | |
| Normalised peak GRF (BW) | −0.055 | 0.024 | 0.026 | 0.111 | 0.070 | 0.117 | 1.34 | 0.535 | 0.014 | 0.172 |
| Normalised peak GRF (BW)* | | | | −0.045 | 0.013 | 0.001 | 0.142 | 0.068 | 0.040 | 0.128 |
| Peak vertical GRF (N) | −11.5 | 5.31 | 0.032 | 47.9 | 15.4 | 0.002 | 281 | 117 | 0.019 | 0.332 |
| Peak vertical GRF (N)* | | | | 15.0 | 2.76 | 0.000 | 28.4 | 14.9 | 0.060 | 0.299 |
| Peak braking GRF (N) | 4.87 | 2.04 | 0.019 | −14.8 | 5.90 | 0.014 | −130 | 45.0 | 0.005 | 0.218 |
| Peak braking GRF (N)* | | | | −0.917 | 1.07 | 0.392 | −23.2 | 5.76 | 0.000 | 0.172 |
| Peak propulsive GRF (N) | | | | 2.03 | 0.818 | 0.015 | 21.5 | 4.42 | 0.000 | 0.285 |
| CoP location at midstance (mm) | | | | −0.123 | 1.54 | 0.937 | −15.2 | 8.34 | 0.071 | 0.036 |
| CoP location at midstance corrected for phalanx size (mm) | | | | −0.437 | 17.0 | 0.980 | −62.7 | 91.9 | 0.497 | 0.005 |

**Appendix 1—table 3.** Crouch factor (CF) and kinematics multiple linear regression results as slopes, standard errors, and p-values.

Models with a significant interaction are displayed in full, and as a simplified model without the interaction term included (marked *). The fit of the model is represented by $R^2$ and relationships are considered significant at $p<0.05$.

| | Interaction | | | Body mass | | | Speed | | | $R^2$ |
|---|---|---|---|---|---|---|---|---|---|---|
| | β | SE | P | β | SE | P | β | SE | P | |
| Maximum CF | | | | –2.19 | 1.07 | 0.043 | –15.8 | 5.74 | 0.007 | 0.122 |
| Minimum CF | | | | –2.45 | 0.885 | 0.007 | –1.53 | 4.76 | 0.749 | 0.064 |
| Change in CF | | | | 0.256 | 0.661 | 0.699 | –14.2 | 3.55 | 0.000 | 0.129 |
| Pelvis pitch ROM (deg) | 0.935 | 0.298 | 0.002 | –2.60 | 0.862 | 0.003 | –19.9 | 6.58 | 0.003 | 0.100 |
| Pelvis pitch ROM (deg)* | | | | 0.066 | 0.159 | 0.677 | 0.534 | 0.858 | 0.535 | 0.008 |
| Hip ROM (deg) | | | | –0.488 | 0.212 | 0.024 | 0.490 | 1.15 | 0.670 | 0.052 |
| Minimum hip flexion (deg) | | | | –0.655 | 0.315 | 0.040 | 0.703 | 1.70 | 0.681 | 0.043 |
| Maximum hip flexion (deg) | | | | –0.167 | 0.260 | 0.520 | 0.213 | 1.40 | 0.880 | 0.004 |
| Knee ROM (deg) | –1.31 | 0.520 | 0.013 | 2.88 | 1.50 | 0.058 | 28.6 | 11.5 | 0.014 | 0.154 |
| Knee ROM (deg)* | | | | –0.850 | 0.272 | 0.002 | –0.120 | 1.47 | 0.935 | 0.098 |
| Minimum knee flexion (deg) | –1.72 | 0.649 | 0.010 | 3.76 | 1.88 | 0.048 | 39.5 | 14.3 | 0.007 | 0.162 |
| Minimum knee flexion (deg)* | | | | –1.13 | 0.34 | 0.001 | 1.95 | 1.84 | 0.294 | 0.101 |
| Maximum knee flexion (deg) | | | | –0.277 | 0.265 | 0.299 | 2.07 | 1.43 | 0.152 | 0.026 |
| Ankle ROM (deg) | | | | 1.02 | 0.166 | 0.000 | 1.63 | 0.899 | 0.073 | 0.339 |
| Peak ankle plantarflexion (deg) | | | | –0.173 | 0.240 | 0.474 | –3.87 | 1.30 | 0.004 | 0.104 |
| Peak ankle dorsiflexion (deg) | | | | –1.19 | 0.198 | 0.000 | –5.49 | 1.07 | 0.000 | 0.466 |
| MTP ROM (deg) | | | | –0.098 | 0.259 | 0.707 | 7.16 | 1.40 | 0.000 | 0.219 |
| Peak MTP plantarflexion (deg) | | | | 0.875 | 0.306 | 0.005 | 5.64 | 1.65 | 0.001 | 0.216 |
| Peak MTP dorsiflexion (deg) | | | | 0.973 | 0.221 | 0.000 | –1.52 | 1.20 | 0.205 | 0.166 |

**Appendix 1—table 4.** Multiple linear regression results of dimensionless peak joint moments as slopes, standard errors, and p-values.

Models with a significant interaction are displayed in full, and as a simplified model without the interaction term included (marked *). The fit of the model is represented by $R^2$ and relationships are considered significant at $p<0.05$.

| | Interaction | | | Body mass | | | Speed | | | $R^2$ |
|---|---|---|---|---|---|---|---|---|---|---|
| | β | SE | P | β | SE | P | β | SE | P | |
| Hip extensor moment | –0.020 | 0.006 | 0.001 | 0.049 | 0.018 | 0.007 | 0.525 | 0.136 | 0.000 | 0.268 |
| Hip extensor moment* | | | | –0.009 | 0.003 | 0.010 | 0.079 | 0.018 | 0.000 | 0.184 |
| Knee extensor moment | | | | 0.000 | 0.001 | 0.766 | 0.003 | 0.008 | 0.721 | 0.003 |
| Knee flexor moment | 0.020 | 0.005 | 0.000 | –0.048 | 0.015 | 0.002 | –0.487 | 0.116 | 0.000 | 0.264 |
| Knee flexor moment* | | | | 0.008 | 0.003 | 0.007 | –0.060 | 0.015 | 0.000 | 0.158 |
| Ankle extensor moment | | | | –0.004 | 0.004 | 0.256 | 0.043 | 0.020 | 0.036 | 0.048 |
| MTP extensor moment | | | | –0.001 | 0.002 | 0.767 | 0.000 | 0.013 | 0.986 | 0.001 |

**Appendix 1—table 5.** Tendon stress and effective mechanical advantage (EMA) multiple linear regression results as slopes, standard errors, and p-values.

Models with a significant interaction are displayed in full, and as a simplified model without the interaction term included (marked *). The fit of the model is represented by $R^2$ and relationships are considered significant at $p<0.05$.

| | Interaction | | | Body mass | | | Speed | | | $R^2$ |
|---|---|---|---|---|---|---|---|---|---|---|
| | β | SE | P | β | SE | P | β | SE | P | |
| *r* at midstance (mm) | | | | −0.063 | 0.088 | 0.477 | −1.88 | 0.474 | 0.000 | 0.173 |
| *R* at midstance (mm) | | | | 3.49 | 1.03 | 0.001 | −0.93 | 5.59 | 0.869 | 0.117 |
| EMA at midstance | | | | −6.64 | 1.98 | 0.001 | −11.0 | 10.7 | 0.310 | 0.142 |
| Peak tendon stress (MPa) | | | | 1.03 | 0.385 | 0.008 | 5.61 | 2.08 | 0.008 | 0.168 |
| Normalised peak tendon stress | | | | −0.048 | 0.018 | 0.008 | 0.258 | 0.096 | 0.008 | 0.107 |
| Peak tendon stress timing % stance | | | | −0.151 | 0.154 | 0.328 | −3.13 | 0.831 | 0.000 | 0.159 |
| Ankle height from ground (mm) | | | | −0.298 | 0.706 | 0.674 | −23.0 | 3.81 | 0.000 | 0.295 |

**Appendix 1—table 6.** Joint net positive, negative and net work simple linear regression (lm(joint work ~mass), lm(joint work ~speed)) results as slopes, standard errors, and p-values.

Work is normalised by body mass (BM). The fit of the model is represented by $R^2$ and relationships are considered significant at $p<0.05$.

| | Body mass | | | | Speed | | | |
|---|---|---|---|---|---|---|---|---|
| | β | SE | P | $R^2$ | β | SE | P | $R^2$ |
| Hip pos | −0.010 | 0.008 | 0.238 | 0.014 | 0.030 | 0.044 | 0.497 | 0.005 |
| Hip neg | −0.027 | 0.005 | 0.000 | 0.225 | −0.058 | 0.031 | 0.066 | 0.034 |
| Hip net | 0.018 | 0.007 | 0.018 | 0.055 | 0.087 | 0.041 | 0.034 | 0.045 |
| Knee pos | −0.021 | 0.011 | 0.052 | 0.038 | 0.145 | 0.057 | 0.013 | 0.061 |
| Knee neg | −0.004 | 0.004 | 0.267 | 0.013 | −0.046 | 0.020 | 0.026 | 0.050 |
| Knee net | −0.017 | 0.012 | 0.169 | 0.019 | 0.191 | 0.064 | 0.003 | 0.084 |
| Ankle pos | 0.059 | 0.019 | 0.003 | 0.087 | 0.478 | 0.097 | 0.000 | 0.197 |
| Ankle neg | 0.022 | 0.017 | 0.200 | 0.017 | 0.321 | 0.087 | 0.000 | 0.122 |
| Ankle net | 0.037 | 0.017 | 0.037 | 0.044 | 0.156 | 0.095 | 0.102 | 0.027 |
| MTP pos | −0.003 | 0.008 | 0.743 | 0.001 | −0.046 | 0.043 | 0.286 | 0.012 |
| MTP neg | 0.023 | 0.014 | 0.089 | 0.029 | 0.149 | 0.073 | 0.044 | 0.041 |
| MTP net | −0.026 | 0.012 | 0.035 | 0.045 | −0.194 | 0.064 | 0.003 | 0.086 |

**Appendix 1—table 7.** Positive, negative and net joint work multiple linear regression results as slopes, standard errors, and p-values.

Work is normalised by body mass (BM). Models with a significant interaction are displayed in full, and as a simplified model without the interaction term included (marked *). The fit of the model is represented by $R^2$ and relationships are considered significant at $p<0.05$.

| Joint work (Jkg$^{-1}$ BM) | Interaction | | | Body mass | | | Speed | | | $R^2$ |
|---|---|---|---|---|---|---|---|---|---|---|
| | β | SE | P | β | SE | P | β | SE | P | |
| Hip positive | | | | 0.046 | 0.045 | 0.307 | −0.012 | 0.008 | 0.161 | 0.025 |
| Hip negative | | | | −0.021 | 0.029 | 0.472 | −0.026 | 0.005 | 0.000 | 0.229 |
| Hip net | | | | 0.067 | 0.042 | 0.110 | 0.015 | 0.008 | 0.058 | 0.080 |
| Knee positive | −0.072 | 0.020 | 0.000 | 1.766 | 0.431 | 0.000 | 0.175 | 0.057 | 0.003 | 0.241 |

*Appendix 1—table 7 Continued on next page*

*Appendix 1—table 7 Continued*

| Joint work (Jkg⁻¹ BM) | Interaction | | | Body mass | | | Speed | | | R² |
|---|---|---|---|---|---|---|---|---|---|---|
| Knee positive* | | | | 0.187 | 0.057 | 0.002 | –0.030 | 0.011 | 0.006 | 0.133 |
| Knee negative | | | | –0.043 | 0.021 | 0.045 | –0.002 | 0.004 | 0.570 | 0.053 |
| Knee net | –0.073 | 0.022 | 0.002 | 1.818 | 0.491 | 0.000 | 0.179 | 0.064 | 0.007 | 0.219 |
| Knee net* | | | | 0.230 | 0.064 | 0.001 | –0.028 | 0.012 | 0.021 | 0.133 |
| Ankle positive | | | | 0.424 | 0.099 | 0.000 | 0.039 | 0.018 | 0.038 | 0.232 |
| Ankle negative | –0.080 | 0.032 | 0.015 | 2.052 | 0.707 | 0.005 | 0.234 | 0.093 | 0.013 | 0.176 |
| Ankle negative* | | | | 0.311 | 0.091 | 0.001 | 0.007 | 0.017 | 0.671 | 0.123 |
| Ankle net | 0.122 | 0.033 | 0.000 | –2.551 | 0.732 | 0.001 | –0.315 | 0.096 | 0.001 | 0.173 |
| Ankle net* | | | | 0.112 | 0.097 | 0.250 | 0.031 | 0.018 | 0.084 | 0.057 |
| MTP positive | | | | –0.045 | 0.044 | 0.312 | 0.000 | 0.008 | 0.956 | 0.012 |
| MTP negative | | | | 0.125 | 0.075 | 0.101 | 0.017 | 0.014 | 0.217 | 0.056 |
| MTP net | | | | –0.170 | 0.066 | 0.011 | –0.018 | 0.012 | 0.149 | 0.106 |

**Appendix 1—table 8.** The mean and standard deviation of joint work for all trials. Positive, negative, and net work is presented for each joint.

| Joint work (Jkg⁻¹ BM) | Mean | SD |
|---|---|---|
| Hip positive | 0.38 | 0.246 |
| Hip negative | 0.187 | 0.178 |
| Hip net | 0.193 | 0.235 |
| Knee positive | 0.432 | 0.334 |
| Knee negative | 0.104 | 0.117 |
| Knee net | 0.328 | 0.375 |
| Ankle positive | 1.99 | 0.613 |
| Ankle negative | 1.324 | 0.525 |
| Ankle net | 0.666 | 0.542 |
| MTP positive | 0.294 | 0.242 |
| MTP negative | 0.652 | 0.42 |
| MTP net | –0.358 | 0.377 |

