## [Editor Report · eLife Assessment]

This **valuable** biomechanical analysis of kangaroo kinematics and kinetics across a range of hopping speeds and masses is a step towards understanding a long-standing problem in locomotion biomechanics: the mechanism for how kangaroos, unlike other mammals, can increase hopping speed without a concomitant increase in metabolic cost. The authors **convincingly** demonstrate that changes in kangaroo posture with speed increase tendon stress/strain and hence elastic energy storage/return. This greater tendon elastic energy storage/return may counteract the increased cost of generating muscular force at faster speeds and thus allows for the invariance in metabolic cost. This methodologically impressive study sets the stage for further work to investigate the relation of hopping speed to metabolic cost more definitively.

---

## [Referee Report · Reviewer #1 (Public review)]

Summary:

The study explored the biomechanics of kangaroo hopping across both speed and animal size to try and explain the unique and remarkable energetics of kangaroo locomotion.

Strengths:

Brings kangaroo locomotion biomechanics into the 21st century. Remarkably difficult project to accomplish. Excellent attention to detail. Clear writing and figures.

General Comments

This is a very impressive tour de force by an all-star collaborative team of researchers. The study represents a tremendous leap forward (pun intended) in terms of our understanding of kangaroo locomotion. Some might wonder why such an unusual species is of much interest. But, in my opinion, the classic study by Dawson and Taylor in 1973 of kangaroos launched the modern era of running biomechanics/energetics and applies to varying degrees to all animals that use bouncing gaits (running, trotting, galloping and of course hopping). The puzzling metabolic energetics findings of Dawson & Taylor (little if any increase in metabolic power despite increasing forward speed) remain a giant unsolved problem in comparative locomotor biomechanics and energetics. It is our "dark matter problem".

This study is certainly a hop towards solving the problem. The study clearly shows that the ankle and to a lesser extent the mtp joint are where the action is. They show in great detail by how much and by what means the ankle joint tendons experience increased stress at faster forward speeds. Since these were zoo animals, direct measures were not feasible, but the conclusion that the tendons are storing and returning more elastic energy per hop at faster speeds is solid. The conclusion that net muscle work per hop changes little from slow to fast forward speeds is also solid. Doing less muscle work can only be good if one is trying to minimize metabolic energy consumption. However, to achieve the greater tendon stresses, there must be greater muscle forces. Unless one is willing to reject the premise of the cost of generating force hypothesis, that is an important issue to confront. Further, the present data support the Kram & Dawson finding of decreased contact times at faster forward speeds. Kram & Taylor and subsequent applications of (and challenges to) their approach support the idea that shorter contact times (tc) require recruiting more expensive muscle fibers and hence greater metabolic costs. The present authors have clarified that this study has still not tied up the metabolic energetics across speed problem and they now point out how the group is now uniquely and enviably poised to explore the problem more using a dynamic SIMM model that incorporates muscle energetics.

---

## [Author Response]

The following is the authors’ response to the original reviews.

**Reviewer #1 (Public Review):**
Summary:The study explored the biomechanics of kangaroo hopping across both speed and animal size to try and explain the unique and remarkable energetics of kangaroo locomotion.Strengths:The study brings kangaroo locomotion biomechanics into the 21st century. It is a remarkably difficult project to accomplish. There is excellent attention to detail, supported by clear writing and figures.Weaknesses:The authors oversell their findings, but the mystery still persists.The manuscript lacks a big-picture summary with pointers to how one might resolve the big question.General CommentsThis is a very impressive tour de force by an all-star collaborative team of researchers. The study represents a tremendous leap forward (pun intended) in terms of our understanding of kangaroo locomotion. Some might wonder why such an unusual species is of much interest. But, in my opinion, the classic study by Dawson and Taylor in 1973 of kangaroos launched the modern era of running biomechanics/energetics and applies to varying degrees to all animals that use bouncing gaits (running, trotting, galloping and of course hopping). The puzzling metabolic energetics findings of Dawson & Taylor (little if any increase in metabolic power despite increasing forward speed) remain a giant unsolved problem in comparative locomotor biomechanics and energetics. It is our "dark matter problem".

Thank you for the kind words.

This study is certainly a hop towards solving the problem. But, the title of the paper overpromises and the authors present little attempt to provide an overview of the remaining big issues.

We have modified the title to reflect this comment. “Postural adaptations may contribute to the unique locomotor energetics seen in hopping kangaroos”

The study clearly shows that the ankle and to a lesser extent the mtp joint are where the action is. They clearly show in great detail by how much and by what means the ankle joint tendons experience increased stress at faster forward speeds.Since these were zoo animals, direct measures were not feasible, but the conclusion that the tendons are storing and returning more elastic energy per hop at faster speeds is solid. The conclusion that net muscle work per hop changes little from slow to fast forward speeds is also solid.Doing less muscle work can only be good if one is trying to minimize metabolic energy consumption. However, to achieve greater tendon stresses, there must be greater muscle forces. Unless one is willing to reject the premise of the cost of generating force hypothesis, that is an important issue to confront. Further, the present data support the Kram & Dawson finding of decreased contact times at faster forward speeds. Kram & Taylor and subsequent applications of (and challenges to) their approach supports the idea that shorter contact times (tc) require recruiting more expensive muscle fibers and hence greater metabolic costs. Therefore, I think that it is incumbent on the present authors to clarify that this study has still not tied up the metabolic energetics across speed problems and placed a bow atop the package.Fortunately, I am confident that the impressive collective brain power that comprises this author list can craft a paragraph or two that summarizes these ideas and points out how the group is now uniquely and enviably poised to explore the problem more using a dynamic SIMM model that incorporates muscle energetics (perhaps ala' Umberger et al.). Or perhaps they have other ideas about how they can really solve the problem.

You have raised important points, thank you for this feedback. We have added a limitations and considerations section to the discussion which highlights that there are still unanswered questions. Line 311-328

Considerations and limitations

“First, we believe it is more likely that the changes in moment arms and EMA can be attributed to speed rather than body mass, given the marked changes in joint angles and ankle height observed at faster hopping speeds. However, our sample included a relatively narrow range of body masses (13.7 to 26.6 kg) compared to the potential range (up to 80 kg), limiting our ability to entirely isolate the effects of speed from those of mass. Future work should examine a broader range of body sizes. Second, kangaroos studied here only hopped at relatively slow speeds, which bounds our estimates of EMA and tendon stress to a less critical region. As such, we were unable to assess tendon stress at fast speeds, where increased forces would reduce tendon safety factors closer to failure. A different experimental or modelling approach may be needed, as kangaroos in enclosures seem unwilling to hop faster over force plates. Finally, we did not determine whether the EMA of proximal hindlimb joints (which are more difficult to track via surface motion capture markers) remained constant with speed. Although the hip and knee contribute substantially less work than the ankle joint (Fig. 4), the majority of kangaroo skeletal muscle is located around these proximal joints. A change in EMA at the hip or knee could influence a larger muscle mass than at the ankle, potentially counteracting or enhancing energy savings in the ankle extensor muscle-tendon units. Further research is needed to understand how posture and muscles throughout the whole body contribute to kangaroo energetics.”

Additionally, we added a line “Peak GRF also naturally increased with speed together with shorter ground contact durations (Fig. 2b, Suppl. Fig 1b)” (line 238) to highlight that we are not proposing that changes in EMA alone explain the full increase in tendon stress. Both GRF and EMA contribute substantially (almost equally) to stress, and we now give more equal discussion to both. For instance, we now also evaluate how much each contributes: “If peak GRF were constant but EMA changed from the average value of a slow hop to a fast hop, then stress would increase 18%, whereas if EMA remained constant and GRF varied by the same principles, then stress would only increase by 12%. Thus, changing posture and decreasing ground contact duration both appear to influence tendon stress for kangaroos, at least for the range of speeds we examined” (Line 245-249)

We have added a paragraph in the discussion acknowledging that the cost of generating force problem is not resolved by our work, concluding that “This mechanism may help explain why hopping macropods do not follow the energetic trends observed in other species (Dawson and Taylor 1973, Baudinette et al. 1992, Kram and Dawson 1998), but it does not fully resolve the cost of generating force conundrum” Line 274-276.

I have a few issues with the other half of this study (i.e. animal size effects). I would enjoy reading a new paragraph by these authors in the Discussion that considers the evolutionary origins and implications of such small safety factors. Surely, it would need to be speculative, but that's OK.

We appreciate this comment from the reviewer, however could not extend the study to discuss animal size effects because, as we now note in the results: “The range of body masses may not be sufficient to detect an effect of mass on ankle moment in addition to the effect of speed.” Line 193

**Reviewer #2 (Public Review):**
SummaryThis is a fascinating topic that has intrigued scientists for decades. I applaud the authors for trying to tackle this enigma. In this manuscript, the authors primarily measured hopping biomechanics data from kangaroos and performed inverse dynamics.While these biomechanical analyses were thorough and impressively incorporated collected anatomical data and an Opensim model, I'm afraid that they did not satisfactorily address how kangaroos can hop faster and not consume more metabolic energy, unique from other animals. Noticeably, the authors did not collect metabolic data nor did they model metabolic rates using their modelling framework. Instead, they performed a somewhat traditional inverse dynamics analysis from multiple animals hopping at a self-selected speed.

In the current study, we aimed to provide a joint-level explanation for the increases of tendon stress that are likely linked to metabolic energy consumption.

We have now included a limitations section in the manuscript (See response to Rev 1). We plan to expand upon muscle level energetics in the future with a more detailed musculoskeletal model.

Within these analyses, the authors largely focused on ankle EMA, discussing its potential importance (because it affects tendon stress, which affects tendon strain energy, which affects muscle mechanics) on the metabolic cost of hopping. However, EMA was roughly estimated (CoP was fixed to the foot, not measured) and did not detectibly associate with hopping speed see results Yet, the authors interpret their EMA findings as though it systematically related with speed to explain their theory on how metabolic cost is unique in kangaroos vs. other animals

As noted in our methods, EMA was not calculated from a fixed centre of pressure (CoP). We did fix the medial-lateral position, owing to the fact that both feet contacted the force plate together, but the anteroposterior movement of the CoP was recorded by the force plate and thus allowed to move. We report the movement (or lack of movement) in our results. The anterior-posterior axis is the most relevant to lengthening or shortening the distance of the ‘out-lever’ R, and thereby EMA. It is necessary to assume fixed medial-lateral position because a single force trace and CoP is recorded when two feet land on the force plate. The mediallateral forces on each foot cancel out so there is no overall medial-lateral movement if the forces are symmetrical (e.g. if the kangaroo is hopping in a straight path and one foot is not in front of the other). We only used symmetrical trials so that the anterior-posterior movement of the CoP would be reliable. We have now added additional details into the text to clarify this

Indeed, the relationship between R and speed (and therefore EMA and speed) was not significant. However, the significant change in ankle height with speed, combined with no systematic change in COP at midstance, demonstrates that R would be greater at faster speeds. If we consider the nonsignificant relationship between R and speed to indicate that there is no change in R, then these two results conflict. We could not find a flaw in our methods, so instead concluded that the nonsignificant relationship between R and speed may be due to a small change in R being undetectable in our data. Taking both results into account, we believe it is more likely that there is a non-detectable change in R, rather than no change in R with speed, but we presented both results for transparency. We have added an additional section into the results to make this clearer (Line 177-185) “If we consider the nonsignificant relationship between R (and EMA) and speed to indicate that there is no change in R, then it conflicts with the ankle height and CoP result. Taking both into account, we think it is more likely that there is a small, but important, change in R, rather than no change in R with speed. It may be undetectable because we expect small effect sizes compared to the measurement range and measurement error (Suppl. Fig. 3h), or be obscured by a similar change in R with body mass. R is highly dependent on the length of the metatarsal segment, which is longer in larger kangaroos (1 kg BM corresponded to ~1% longer segment, P<0.001, R^2^=0.449). If R does indeed increase with speed, both R and r will tend to decrease EMA at faster speeds.”

These speed vs. biomechanics relationships were limited by comparisons across different animals hopping at different speeds and could have been strengthened using repeated measures design

There is significant variation in speed within individuals, not just between individuals. The preferred speed of kangaroos is 2-4.5 m/s, but most individuals showed a wide speed range within this. Eight of our 16 kangaroos had a maximum speed that was 1-2m/s faster than their slowest trial. Repeated measures of these eight individuals comprises 78 out of the 100 trials. It would be ideal to collect data across the full range of speeds for all individuals, but it is not feasible in this type of experimental setting. Interference with animals such as chasing is dangerous to kangaroos as they are prone to adverse reactions to stress. We have now added additional information about the chosen hopping speeds into the results and methods sections to clarify this “The kangaroos elected to hop between 1.99 and 4.48 m s^-1^, with a range of speeds and number of trials for each individual (Suppl. Fig. 9).” (Line 381-382)

There are also multiple inconsistencies between the authors' theory on how mechanics affect energetics and the cited literature, which leaves me somewhat confused and wanting more clarification and information on how mechanics and energetics relate

We thank the reviewer for this comment. Upon rereading we now understand the reviewers position, and have made substantial revisions to the introduction and discussion (See comments below)

My apologies for the less-than-favorable review, I think that this is a neat biomechanics study - but am unsure if it adds much to the literature on the topic of kangaroo hopping energetics in its current form.

Again we thank the reviewer for their time and appreciate their efforts to strengthen our manuscript.

**Reviewer #3 (Public Review):**
Summary:The goal of this study is to understand how, unlike other mammals, kangaroos are able to increase hopping speed without a concomitant increase in metabolic cost. They use a biomechanical analysis of kangaroo hopping data across a range of speeds to investigate how posture, effective mechanical advantage, and tendon stress vary with speed and mass. The main finding is that a change in posture leads to increasing effective mechanical advantage with speed, which ultimately increases tendon elastic energy storage and returns via greater tendon strain. Thus kangaroos may be able to conserve energy with increasing speed by flexing more, which increases tendon strain.Strengths:The approach and effort invested into collecting this valuable dataset of kangaroo locomotion is impressive. The dataset alone is a valuable contribution.

Thank you!

Weaknesses:Despite these strengths, I have concerns regarding the strength of the results and the overall clarity of the paper and methods used (which likely influences how convincingly the main results come across).(1) The paper seems to hinge on the finding that EMA decreases with increasing speed and that this contributes significantly to greater tendon strain estimated with increasing speed. It is very difficult to be convinced by this result for a number of reasons:It appears that kangaroos hopped at their preferred speed. Thus the variability observed is across individuals not within. Is this large enough of a range (either within or across subjects) to make conclusions about the effect of speed, without results being susceptible to differences between subjects?

Apologies, this was not clear in the manuscript. Kangaroos hopping at their preferred speed means we did not chase or startle them into high speeds to comply with ethics and enclosure limitations. Thus we did not record a wide range of speeds within the bounds of what kangaroos are capable of in the wild (up to 12 m/s), but for the range we did measure (~2-4.5 m/s), there is a large amount of variation in hopping speed within each individual kangaroo. Out of 16 individuals, eight individuals had a difference of 1-2m/s between their slowest and fastest trials, and these kangaroos accounted for 78 out of 100 trials. Of the remainder, six individuals had three for fewer trials each, and two individuals had highly repeatable speeds (3 out of 4, and 6 out of 7 trials were within 0.5 m/s). We have now removed the terminology “preferred speed” e.g line 115. We have added additional information about the chosen hopping speeds into the results and methods, including an appendix figure “The kangaroos elected to hop between 1.99 and 4.48 m s^-1^, with a range of speeds and number of trials for each individual (Suppl. Fig. 9).” (Line 381-382)

In the literature cited, what was the range of speeds measured, and was it within or between subjects?

For other literature, to our knowledge the highest speed measured is ~9.5m/s (see supplementary Fig1b) and there were multiple measures for several individuals (see methods Kram & Dawson 1998).

Assuming that there is a compelling relationship between EMA and velocity, how reasonable is it to extrapolate to the conclusion that this increases tendon strain and ultimately saves metabolic cost? They correlate EMA with tendon strain, but this would still not suggest a causal relationship (incidentally the p-value for the correlation is not reported).

The functions that underpin these results (e.g. moment = GRF*R) come from physical mechanics and geometry, rather than statistical correlations. Additionally, a p-value is not appropriate in the relationship between EMA and stress (rather than strain) because the relationship does not appear to be linear. We have made it clearer in the discussion that we are not proposing that entire change in stress is caused by changes in EMA, but that the increase in GRF that naturally occurs with speed will also explain some of the increase in stress, along with other potential mechanisms. The discussion has been extensively revised to reflect this.

Tendon strain could be increasing with ground reaction force, independent of EMA. Even if there is a correlation between strain and EMA, is it not a mathematical necessity in their model that all else being equal, tendon stress will increase as ema decreases? I may be missing something, but nonetheless, it would be helpful for the authors to clarify the strength of the evidence supporting their conclusions.

Yes, GRF also contributes to the increase in tendon stress in the mechanism we propose (Suppl. Fig. 8), see the formulas in Fig 6, and we have made this clearer in the revised discussion (see above comment). You are correct that mathematically stress is inversely proportional to EMA, which can be observed in Fig. 7a, and we did find that EMA decreases.

The statistical approach is not well-described. It is not clear what the form of the statistical model used was and whether the analysis treated each trial individually or grouped trials by the kangaroo. There is also no mention of how many trials per kangaroo, or the range of speeds (or masses) tested.

The methods include the statistical model with the variables that we used, as well as the kangaroo masses (13.7 to 26.6 kg, mean: 20.9 ± 3.4 kg). We did not have sufficient within individual sample size to use a linear mixed effect model including subject as a random factor, thus all trials were treated individually. We have included this information in the results section.

We have now moved the range of speeds from the supplementary material to the results and figure captions. We have added information on the number of trials per kangaroo to the methods, and added Suppl. Fig. 9 showing the distribution of speeds per kangaroo.

We did not group the data e.g. by using an average speed per individual for all their trials, or by comparing fast to slow groups for statistical analysis (the latter was only for display purposes in our figures, which we have now made clearer in the methods statistics section).

Related to this, there is no mention of how different speeds were obtained. It seems that kangaroos hopped at a self-selected pace, thus it appears that not much variation was observed. I appreciate the difficulty of conducting these experiments in a controlled manner, but this doesn’t exempt the authors from providing the details of their approach.

Apologies, this was not clear in the manuscript. Kangaroos hopping at their preferred speed means we did not chase or startle them into high speeds to comply with ethics and enclosure limitations. Thus we did not record a wide range of speeds within the bounds of what kangaroos are capable of in the wild (up to 12 m/s). We have now removed the terminology “preferred speed” e.g. line 115. We have added additional information about the chosen hopping speeds into the results and methods, including an appendix figure (see above comment). (Line 381-382)

Some figures (Figure 2 for example) present means for one of three speeds, yet the speeds are not reported (except in the legend) nor how these bins were determined, nor how many trials or kangaroos fit in each bin. A similar comment applies to the mass categories. It would be more convincing if the authors plotted the main metrics vs. speed to illustrate the significant trends they are reporting.

Thank you for this comment. The bins are used only for display purposes and not within the statistical analysis. We have clarified this in the revised manuscript: “The data was grouped into body mass (small 17.6±2.96 kg, medium 21.5±0.74 kg, large 24.0±1.46 kg) and speed (slow 2.52±0.25 m s^-1^, medium 3.11±0.16 m s^-1^, fast 3.79±0.27 m s^-1^) subsets for display purposes only”. (Line 495-497)

(2) The significance of the effects of mass is not clear. The introduction and abstract suggest that the paper is focused on the effect of speed, yet the effects of mass are reported throughout as well, without a clear understanding of the significance. This weakness is further exaggerated by the fact that the details of the subject masses are not reported.

Indeed, the primary aim of our study was to explore the influence of speed, given the uncoupling of energy from hopping speed in kangaroos. We included mass to ensure that the effects of speed were not driven by body mass (i.e.: that larger kangaroos hopped faster). Subject masses were reported in the first paragraph of the methods, albeit some were estimated as outlined in the same paragraph.

(3) The paper needs to be significantly re-written to better incorporate the methods into the results section. Since the results come before the methods, some of the methods must necessarily be described such that the study can be understood at some level without turning to the dedicated methods section. As written, it is very difficult to understand the basis of the approach, analysis, and metrics without turning to the methods.

The methods after the discussion is a requirement of the journal. We have incorporated some methods in the results where necessary but not too repetitive or disruptive, e.g. Fig. 1 caption, and specifying we are only analysing EMA for the ankle joint

**Reviewing Editor (Recommendations For The Authors):**
Below is a list of specific recommendations that the authors could address to improve the eLife assessment:(1) Based on the data presented and the fact that metabolic energy was not measured, the authors should temper their conclusions and statements throughout the manuscript regarding the link between speed and metabolic energy savings. We recommend adding text to the discussion summarizing the strengths and limitations of the evidence provided and suggesting future steps to more conclusively answer this mystery.

There is a significant body of work linking metabolic energy savings to measured increases in tendon stress in macropods. However, the purpose of this paper was to address the unanswered questions about why tendon stress increases. We found that stress did not only increase due to GRF increasing with speed as expected, but also due to novel postural changes which decreased EMA. In the revised manuscript, we have tempered our conclusions to make it clearer that it is not just EMA affecting stress, and added limitations throughout the manuscript (see response to Rev 1).

(2) To provide stronger evidence of a link between speed, mechanics, and metabolic savings the authors can consider estimating metabolic energy expenditure from their OpenSIM model. This is one suggestion, but the authors likely have other, possibly better ideas. Such a model should also be able to explain why the metabolic rate increases with speed during uphill hopping.

Extending the model to provide direct metabolic cost estimates will be the goal of a future paper, however the models does not have detailed muscle characteristics to do this in the formulation presented here. It would be a very large undertaking which is beyond the scope of the current manuscript. As per the comment above, the results of this paper are not reliant on metabolic performance.

(3) The authors attempt to relate the newly quantified hopping biomechanics to previously published metabolic data. However, all reviewers agree that the logic in many instances is not clear or contradictory. Could one potential explanation be that at slow speeds, forces and tendon strain are small, and thus muscle fascicle work is high? Then, with faster speeds, even though the cost of generating isometric force increases, this is offset by the reduction in the metabolic cost of muscular work. The paper could provide stronger support for their hypotheses with a much clearer explanation of how the kinematics relate to the mechanics and ultimately energy savings.

In response to the reviewers comments, we have substantially modified the discussion to provide clearer rationale.

(4) The methods and the effort expended to collect these data are impressive, but there are a number of underlying assumptions made that undermine the conclusions. This is due partly to the methods used, but also the paper's incomplete description of their methods. We provide a few examples below:It would be helpful if the authors could speak to the effect of the limited speeds tested and between-animal comparisons on the ability to draw strong conclusions from the present dataset. ·Throughout the discussion, the authors highlight the relationship between EMA and speed. However, this is misleading since there was no significant effect of speed on EMA. Speed only affected the muscle moment arm, r. At minimum, this should be clarified and the effect on EMA not be overstated. Additionally, the resulting implications on their ability to confidently say something about the effect of speed on muscle stress should be discussed.

We have now provided additional details, (see responses above) to these concerns. For instance, we added a supplementary figure showing the speed distribution per individual. The primary reviewer concern (that each kangaroo travelled at a single speed) was due to a miscommunication around the terminology “preferred” which has now been corrected.

We now elaborate in the results why we are not very concerned that EMA is insignificant. The statistical insignificance of EMA is ultimately due to the insignificance of the direct measurement of R, however, we now better explain in the results why we believe that this statistical insignificance is due to error/noise of the measurement which is relatively large compared to the effect size. Indirect indications of how R may increase with speed (via ankle height from the ground) are statistically significant. Lines 177-185.

We consider this worth reporting because, for instance, an 18% change in EMA will be undetectable by measurement, but corresponds to an 18% change in tendon stress which is measurable and physiologically significant (safety factor would decrease from 2 to 1.67). We presented both significant and insignificant results for transparency.

We have also discussed this within a revised limitations section of the manuscript (Line 311328).

**Reviewer #1 (Recommendations For The Authors):**
Title: I would cut the first half of the title. At least hedge it a bit. "Clues" instead of "Unlocking the secrets".

We have revised the title to: “Postural adaptations may contribute to the unique locomotor energetics seen in hopping kangaroos”

In my comments, ... typically indicates a stylistic change suggested to the text.Overall, the paper covers speed and size. Unfortunately, the authors were not 100% consistent in the order of presenting size then speed, or speed then size. Just choose one and stick with it.

We have attempted to keep the order of presenting size and speed consistent, however there are several cases where this would reduce the readability of the manuscript and so in some cases this may vary.

One must admit that there is a lot of vertical scatter in almost all of the plots. I understand that these animals were not in a lab on a treadmill at a controlled speed and the animals wear fur coats so marker placements vary/move etc. But the spread is quite striking, e.g. Figure 5a the span at one speed is almost 10x. Can the authors address this somewhere? Limitations section?

The variation seen likely results from attempting to display data in a 2D format, when it is in fact the result of multiple variables, including speed, mass, stride frequency and subject specific lengths. Slight variations in these would be expected to produce some noise around the mean, and I think it’s important to consider this while showing the more dominant effects.

In many locations in the manuscript, the term "work" is used, but rarely if ever specified that this is the work "per hop". The big question revolves around the rate of metabolic energy consumption (i.e. energy per time or average metabolic power), one must not forget that hop frequency changes somewhat across speed, so work per hop is not the final calculation.

Thank you for this comment. We have now explicitly stated work per hop in figure captions and in the results (line 208). The change in stride frequency at this range of speeds is very small, particularly compared to the variance in stride frequency (Suppl. Fig. 1d), which is consistent with other researchers who found that stride frequency was constant or near constant in macropods at analogous speeds (e.g. Dawson and Taylor 1973, Baudinette et al. 1987).

Line 61 ....is likely related.

Added “likely” (line 59)

Line 86 I think the Allen reference is incomplete. Wasn't it in J Exp Biology?

Thank you. Changed.

Line 122 ... at faster speeds and in larger individuals.

Changed: “We hypothesised that (i) the hindlimb would be more crouched at faster speeds, primarily due to the distal hindlimb joints (ankle and metatarsophalangeal), independent of changes with body mass” (Line 121-122).

Line 124 I found this confusing. Try to re-word so that you explain you mean more work done by the tendons and less by the ankle musculature.

Amended: “changes in moment arms resulting from the change in posture would contribute to the increase in tendon stress with speed, and may thereby contribute to energetic savings by increasing the amount of positive and negative work done by the ankle without requiring additional muscle work” (Line 123)

Line 129 hopefully "braking" not "breaking"!

Thank you. Fixed. (Line 130)

Line 129 specify fore-aft horizontal force.

Added "fore-aft" to "negative fore-aft horizontal component" (Line 130-131)

Line 130 add something like "of course" or "naturally" since if there is zero fore-aft force, the GRF vector of course must be vertical.

Added "naturally" (Line 132)

Line 138 clarify that this section is all stance phase. I don't recall reading any swing phase data.

Changed to: "Kangaroo hindlimb stance phase kinematics varied…" (Line 141)

Line 143 and elsewhere. I found the use of dorsiflexion and plantarflexion confusing. In Figure 3, I see the ankle never flexing more than 90 degrees. So, the ankle joint is always in something of a flexed position, though of course it flexes and extends during contact. I urge the authors to simplify to flextion/extension and drop the plantar/dorsi.

We have edited this section to describe both movements as greater extension (plantarflexion). (Line 147). We have further clarified this in the figure caption for figure 3.

Line 147 ...changes were…

Fixed, line 150

Line 155 I'm a bit confused here. Are the authors calculating some sort of overall EMA or are they saying all of the individual joint EMAs all decreased?

Thank you, we clarified that it is at the ankle. Line 158

Line 158 since kangaroos hop and are thus positioned high and low throughout the stance phase, try to avoid using "high" and "low" for describing variables, e.g. GRF or other variables. Just use "greater/greatest" etc.

Thanks for this suggestion. We have changed "higher" into "greater" where appropriate throughout the manuscript e.g. line 161

Lines 162 and 168 same comment here about "r" and "R". Do you mean ankle or all joints?

Clarified that it is the gastrocnemius and plantaris r, and the R to the ankle. (Lines 164-165)

Line 173 really, ankle height?

Added: ankle height is "vertical distance from the ground". Line 177

Line 177 is this just the ankle r?

Added "of the ankle" line 158 and “Achilles” line 187

Line 183 same idea, which tendon/tendons are you talking about here?

Added "Achilles" to be more clear (Line 187)

Line 195 substitute "converted" for "transferred".

Done (Line 210)

Line 223 why so vague? i.e. why use "may"? Believe in your data. ...stress was also modulated by changes....

Changed "may" to "is"

Line 229 smaller ankle EMA (especially since you earlier talked about ankle "height").

Changed “lower” to “smaller” Line 254

Line 2236 ...and return elastic energy…

Added "elastic" line 262

Line 244 IMPORTANT: Need to explain this better! I think you are saying that the net work at the ankle is staying the same across speed, BUT it is the tendons that are storing and returning that work, it's not that the muscles are doing a lot of negative/positive work.

Changed: “The consistent net work observed among all speeds suggests the ankle extensor muscle-tendon units are performing similar amounts of ankle work independent of speed, which would predominantly be done by the tendon.” Line 270-272

Line 258-261 I think here is where you are over-selling the data/story. Although you do say "a" mechanism and not "the" mechanism, you still need to deal with the cost of generating more force and generating that force faster.

We removed this sentence and replaced it with a discussion of the cost of generating force hypothesis, and alternative scenarios for the how force and metabolics could be uncoupled.

Line 278 "the" tendon? Which tendon?

Added "Achilles"

Line 289. I don't think one can project into the past.

Changed “projected” to "estimated"

Line 303 no problem, but I've never seen a paper in biology where the authors admit they don't know what species they were studying!

Can’t be helped unfortunately. It is an old dataset and there aren’t photos of every kangaroo. Fortunately, from the grey and red kangaroos we can distinguish between, we know there are no discernible species effects on the data.

Lines 304-306 I'm not clear here. Did you use vertical impulse (and aerial time) to calculate body weight? Or did you somehow use the braking/propulsive impulse to calculate mass? I would have just put some apples on the force plate and waited for them to stop for a snack.

Stationary weights were recorded for some kangaroos which did stand on the force plate long enough, but unfortunately not all of them were willing to do so. In those cases, yes, we used impulse from steady-speed trials to estimate mass. We cross-checked by estimated mass from segment lengths (as size and mass are correlated). This is outlined in the first paragraph of the methods.

Lines 367 & 401 When you use the word "scaled" do you mean you assumed geometric similarity?

No, rather than geometric scaling, we allowed scaling to individual dimensions by using the markers at midstance for measurements. We have amended the paragraph to clarify that the shape of the kangaroo changes and that mass distribution was preserved during the shape change (line 441-446)

Lines 381-82 specify "joint work"

Added "joint work" (Line 457)

Figure 1 is gorgeous. Why not add the CF equation to the left panel of the caption?

We decided to keep the information in the figure caption. “Total leg length was calculated as the sum of the segment lengths (solid black lines) in the hindlimb and compared to the pelvisto-toe distance (dashed line) to calculate the crouch factor”

Figure 2 specify Horizontal fore-aft.

Done

Figure 3g I'd prefer the same Min. Max Flexion vertical axis labels as you use for hip & knee.

While we appreciate the reviewer trying to increase the clarity of this figure, we have left it as plantar/dorsi flexion since these are recognised biomechanical terms. To avoid confusion, we have further defined these in the figure caption “For (f-g), increased plantarflexion represents a decrease in joint flexion, while increased dorsiflexion represents increased flexion of the joint.”

Figure 4. I like it and I think that you scaled all panels the same, i.e. 400 W is represented by the same vertical distance in all panels. But if that's true, please state so in the Caption. It's remarkable how little work occurs at the hip and knee despite the relatively huge muscles there.

Is it true that the y axes are all at the same scale. We have added this to the caption.

Figure 5 Caption should specify "work per hop".

Added

Figure 7 is another beauty.

Thank you!

Supplementary Figure 3 is this all ANKLE? Please specify.

Clarified that it is the gastrocnemius and plantaris r, and the R to the ankle.

**Reviewer #2 (Recommendations For The Authors):**
To 'unlock the secrets of kangaroo locomotor energetics' I expected the authors to measure the secretive outcome variable, metabolic rate using laboratory measures. Rather, the authors relied on reviewing historic metabolic data and collecting biomechanics data across different animals, which limits the conclusions of this manuscript.

We have revised to the title to make it clearer that we are investigating a subset of the energetics problem, specifically posture. “Postural adaptations may contribute to the unique locomotor energetics seen in hopping kangaroos.” We have also substantially modified the discussion to temper the conclusions from the paper.

After reading the hypothesis, why do the authors hypothesize about joint flexion and not EMA? Because the following hypothesis discusses the implications of moment arms on tendon stress, EMA predictions are more relevant (and much more discussed throughout the manuscript).

Ankle and MTP angles are the primary drivers of changes in r, R & thus, EMA. We used a two part hypothesis to capture this. We have rephased the hypotheses: “We hypothesised that (i) the hindlimb would be more crouched at faster speeds, primarily due to the distal hindlimb joints (ankle and metatarsophalangeal), independent of changes with body mass, and (ii) changes in moment arms resulting from the change in posture would contribute to the increase in tendon stress with speed, and may thereby contribute to energetic savings by increasing the amount of positive and negative work done by the ankle without requiring additional muscle work.”

If there were no detectable effects of speed on EMA, are kangaroos mechanically like other animals (Biewener Science 89 & JAP 04) who don't vary EMA across speeds? Despite no detectible effects, the authors state [lines 228-229] "we found larger and faster kangaroos were more crouched, leading to lower ankle EMA". Can the authors explain this inconsistency? Lines 236 "Kangaroos appear to use changes in posture and EMA". I interpret the paper as EMA does not change across speed.

Apologies, we did not sufficiently explain this originally. We now explain in the results our reasoning behind our belief that EMA and R may change with speed. “If we consider the nonsignificant relationship between R (and EMA) and speed to indicate that there is no change in R, then it conflicts with the ankle height and CoP result. Taking both into account, we think it is more likely that there is a small, but important, change in R, rather than no change in R with speed. It may be undetectable because we expect small effect sizes compared to the measurement range and measurement error (Suppl. Fig. 3h), or be obscured by a similar change in R with body mass. R is highly dependent on the length of the metatarsal segment, which is longer in larger kangaroos (1 kg BM corresponded to ~1% longer segment, P<0.001, R^2^=0.449). If R does indeed increase with speed, both R and r will tend to decrease EMA at faster speeds.” (Line 177-185)

Lines 335-339: "We assumed the force was applied along phalanx IV and that there was no medial or lateral movement of the centre of pressure (CoP)". I'm confused, did the authors not measure CoP location with respect to the kangaroo limb? If not, this simple estimation undermines primary results (EMA analyses).

We have changed "The anterior or posterior movement of the CoP was recorded by the force plate" to read: "The fore-aft movement of the CoP was recorded by the force plate within the motion capture coordinate system" (Line 406-407) and added more justification for fixing the CoP movement in the other axis: “It was necessary to assume the CoP was fixed in the mediallateral axis because when two feet land on the force plate, the lateral forces on each foot are not recorded, and indeed cancel if the forces are symmetrical (i.e. if the kangaroo is hopping in a straight path and one foot is not in front of the other). We only used symmetrical trials to ensure reliable measures of the anterior-posterior movement of the CoP.” (Line 408-413)

The introduction makes many assertions about the generalities of locomotion and the relationship between mechanics and energetics. I'm afraid that the authors are selectively choosing references without thoroughly evaluating alternative theories. For example, Taylor, Kram, & others have multiple papers suggesting that decreasing EMA and increasing muscle force (and active muscle volume) increase metabolic costs during terrestrial locomotion. Rather, the authors suggest that decreasing EMA and increasingly high muscle force at faster speeds don't affect energetics unless muscle work increases substantially (paragraph 2)? If I am following correctly, does this theory conflict with active muscle volume ideas that are peppered throughout this manuscript?

Yes, as you point out, the same mechanism does lead to different results in kangaroos vs humans, for instance, but this is not a contradiction. In all species, decreasing EMA will result in an increase in muscle force due to less efficient leverage (i.e. lower EMA) of the muscles, and the muscle-tendon unit will be required to produce more force to balance the joint moment. As a consequence, human muscles activate a greater volume in order for the muscle-tendon unit to increase muscle work and produce enough force. We are proposing that in kangaroos, the increase in work is done by the achilles tendon rather than the muscles. Previous research suggests that macropod ankle muscles contract isometrically or that the fibres do not shorten more at faster speeds i.e. muscle work does not increase with speed. Instead, the additional force seems to come from the tendon storing and subsequently returning more strain energy (indicated by higher stress). We found that the increase in tendon stress comes from higher ground force at faster speeds, and from it adopting a more crouched posture which increases the tendons’ stresses compared to an upright posture for a given speed (think of this as increasing the tendon’s stress capacity). We have substantially revised the discussion to highlight this.

Similarly, does increased gross or net tendon mechanical energy storage & return improve hopping energetics? Would more tendon stress and strain energy storage with a given hysteresis value also dissipate more mechanical energy, requiring leg muscles to produce more net work? Does net or gross muscle work drive metabolic energy consumption?

Based on the cost of generating force hypothesis, we think that gross muscle work would be linked to driving metabolic energy consumption. Our idea here is that the total body work is a product of the work done by the tendon and the muscle combined. If the tendon has the potential to do more work, then the total work can increase without muscle work needing to increase.

The results interpret speed effects on biomechanics, but each kangaroo was only collected at 1 speed. Are inter-animal comparisons enough to satisfy this investigation?

We have added a figure (Suppl Fig 9) to demonstrate the distribution of speed and number of trials per kangaroo. We have also removed "preferred" from the manuscript as this seems to cause confusion. Most kangaroos travelled at a range of “casual” speeds.

Abstract: Can the authors more fully connect the concept of tendon stress and low metabolic rates during hopping across speeds? Surely, tendon mechanics don't directly drive the metabolic cost of hopping, but they affect muscle mechanics to affect energetics.

Amended to: " This phenomenon may be related to greater elastic energy savings due to increasing tendon stress; however, the mechanisms which enable the rise in stress, without additional muscle work remain poorly understood." (Lines 25-27).

The topic sentence in lines 61-63 may be misleading. The ensuing paragraph does not substantiate the topic sentence stating that ankle MTUs decouple speeds and energetics.

We added "likely" to soften the statement. (Line 59)

Lines 84-86: In humans, does more limb flexion and worse EMA necessitate greater active muscle volume? What about muscle contractile dynamics - See recent papers by Sawicki & colleagues that include Hill-type muscle mechanics in active muscle volume estimates.

Added: “Smaller EMA requires greater muscle force to produce a given force on the ground, thereby demanding a greater volume of active muscle, and presumably greater metabolic rates than larger EMA for the same physiology”. (Line 80-82)

Lines 106: can you give the context of what normal tendon safety factors are?

Good idea. Added: "far lower than the typical safety factor of four to eight for mammalian tendons (Ker et al. 1988)." Line 106-107

I thought EMA was relatively stable across speeds as per Biewener [Science & JAP '04]. However the authors gave an example of an elephant to suggest that it is typically inversely related to speed. Can the authors please explain the disconnect and the most appropriate explanation in this paragraph?

Knee EMA in particular changed with speed in Biewener 2004. What is “typical” probably depends on the group of animals studied; e.g., cursorial quadrupedal mammals generally seem to maintain constant EMA, but other groups do not.

These cases are presented to show a range of consequences for changing EMA (usually with mass, but sometimes with speed). We have made several adjustments to the paragraph to make this clearer. Lines 85-93.

The results depend on the modeled internal moment arm (r). How confident are the authors in their little r prediction? Considering complications of joint mechanics in vivo including muscle bulging. Holzer et al. '20 Sci Rep demonstrated that different models of the human Achilles tendon moment arm predict vastly different relationships between the moment arm and joint angle.

Our values for r and EMA closely align with previous papers which measured/calculate these values in kangaroos, such as Kram 1998, and thus we are confident in our interpretation.

This is a misleading results sentence: Small decreases in EMA correspond to a nontrivial increase in tendon stress, for instance, reducing EMA from 0.242 (mean minimum EMA of the slow group) to 0.206 (mean minimum EMA of the fast group) was associated with an ~18% increase in tendon stress. The authors could alternatively say that a ~15% decrease in EMA was associated with an ~18% increase in tendon stress, which seems pretty comparable.

Thank you for pointing this out, it is important that it is made clearer. Although the change in relative magnitude is approximately the same (as it should be), this does not detract from the importance. The "small decrease in EMA" is referring to the absolute values, particularly in respect to the measurement error/noise. The difference is small enough to have been undetectable with other methods used in previous studies. We have amended the sentence to clarify this.

It now reads: “Subtle decreases in EMA which may have been undetected in previous studies correspond to discernible increases in tendon stress. For instance, reducing EMA from 0.242 (mean minimum EMA of the slow group) to 0.206 (mean minimum EMA of the fast group) was associated with an increase in tendon stress from ~50 MPa to ~60 MPa, decreasing safety factor from 2 to 1.67 (where 1 indicates failure), which is both measurable and physiologically significant.” (Line 195-200)

Lines 243-245: "The consistent net work observed among all speeds suggests the ankle extensors are performing similar amounts of ankle work independent of speed." If this is true, and presumably there is greater limb work performed on the center of mass at faster speeds (Donelan, Kram, Kuo), do more proximal leg joints increase work and energy consumption at faster speeds?

The skin over the proximal leg joints (knee and hip) moves too much to get reliable measures of EMA from the ratio of moment arms. This will be pursued in future work when all muscles are incorporated in the model so knee and hip EMA can be determined from muscle force.

We have added limitations and considerations paragraph to the manuscript: “Finally, we did not determine whether the EMA of proximal hindlimb joints (which are more difficult to track via surface motion capture markers) remained constant with speed. Although the hip and knee contribute substantially less work than the ankle joint (Fig. 4), the majority of kangaroo skeletal muscle is located around these proximal joints. A change in EMA at the hip or knee could influence a larger muscle mass than at the ankle, potentially counteracting or enhancing energy savings in the ankle extensor muscle-tendon units. Further research is needed to understand how posture and muscles throughout the whole body contribute to kangaroo energetics.” (Line 321-328)

Lines 245-246: "Previous studies using sonomicrometry have shown that the muscles of tammar wallabies do not shorten considerably during hops, but rather act near-isometrically as a strut" Which muscles? All muscles? Extensors at a single joint?

Added "gastrocnemius and plantaris" Line 164-165

Lines 249-254: "The cost of generating force hypothesis suggests that faster movement speeds require greater rates of muscle force development, and in turn greater cross-bridge cycling rates, driving up metabolic costs (Taylor et al. 1980, Kram and Taylor 1990). The ability for the ankle extensor muscle fibres to remain isometric and produce similar amounts of work at all speeds may help explain why hopping macropods do not follow the energetic trends observed in quadrupedal species." These sentences confuse me. Kram & Taylor's cost of force-generating hypothesis assumes that producing the same average force over shorter contact times increases metabolic rate. How does 'similar muscle work' across all speeds explain the ability of macropods to use unique energetic trends in the cost of force-generating hypothesis context?

Thank you for highlighting this confusion. We have substantially revised the discussion clarify where the mechanisms presented deviate from the cost of generating force hypothesis. Lines 270-309

**Reviewer #3 (Recommendations For The Authors):**
In addition to the points described in the public review, I have additional, related, specific comments:(1) Results: Please refer to the hypotheses in the results, and relate the the findings back to the hypotheses.

We now relate the findings back to the hypotheses

Line 142 “In partial support of hypothesis (i), greater masses and faster speeds were associated with more crouched hindlimb postures (Fig. 3a,c).”.

Lines 205-206: “The increase in tendon stress with speed, facilitated in part by the change in moment arms by the shift in posture, may explain changes in ankle work (c.f. Hypothesis (ii)).”

(2) Results: please provide the main statistical results either in-line or in a table in the main text.

We (the co-authors) have discussed this at length, and have agreed that the manuscript is far more readable in the format whereby most statistics lie within the supplementary tables, otherwise a reader is met with a wall of statistics. We only include values in the main text when the magnitude is relevant to the arguments presented in the results and discussion.

(3) Line 140: Describe how 'crouched' was defined.

We have now added a brief definition of ‘Crouch factor’ after the figure caption. (Line 143) (Fig. 3a,c; where crouch factor is the ratio of total limb length to pelvis to toe distance).

(4) Line 162: This seems to be a main finding and should be a figure in the main text not supplemental. Additionally, Supplementary Figures 3a and b do not show this finding convincingly There should be a figure plotting r vs speed and r vs mass.

The combination of r and R are represented in the EMA plot in the main text. The r and R plots are relegated to the supplementary because the main text is already very crowded. Thank you for the suggestion for the figure plotting r and R versus speed, this is now included as Suppl. Fig. 3h

(5) Line 166: Supplementary Figure 3g does not show the range of dorsiflexion angles as a function of speed. It shows r vs dorsiflexion angle. Please correct.

Thanks for noticing this, it was supposed to reference Fig 3g rather than Suppl Fig 3g in the sentence regarding speed. We have fixed this, Line 170.

We had added a reference to Suppl Fig 3 on Line 169 as this shows where the peak in r with ankle angle occurs (114.4 degrees).

(6) Line 184: Where are the statistical results for this statement?

The relationship between stress and EMA does not appear to be linear, thus we only present R^^^2 for the power relationship rather than a p-value.

(7) Line 192: The authors should explain how joint work and power relate/support the overall hypotheses. This section also refers to Figures 4 and 5 even though Figures 6 and 7 have already been described. Please reorganize.

We have added a sentence at the end of the work and power section to mention hypothesis (ii) and lead into the discussion where it is elaborated upon.

“The increase in positive and negative ankle work may be due to the increase in tendon stress rather than additional muscle work.” Line 219-220 We have rearranged the figure order.

(8) The statistics are not reported in the main text, but in the supplementary tables. If a result is reported in the main text, please report either in-line or with a table in the main text.

We leave most statistics in the supplementary tables to preserve the readability of the manuscript. We only include values in the main text when the magnitude is relevant to the arguments raised in the results and discussion.